# Learning few-shot imitation as cultural transmission

**Avishkar Bhoopchand** [1], **Bethanie Brownfield**[1], **Adrian Collister**[1],
**Agustin Dal Lago** [1], **Ashley Edwards**[1], **Richard Everett** [1], **Alexandre Fréchette**[1],
**Yanko Gitahy Oliveira**[1], **Edward Hughes** [1] ✉, **Kory W. Mathewson**[1],
**Piermaria Mendolicchio**[1], **Julia Pawar**[1], **Miruna Pîslar**[1], **Alex Platonov**[1],
**Evan Senter**[1], **Sukhdeep Singh**[1], **Alexander Zacherl**[1] & **Lei M. Zhang**[1]

Cultural transmission is the domain-general social skill that allows agents to acquire and use information from each other in real-time with high fidelity and recall. It can be thought of as the process that perpetuates fit variants in cultural evolution. In humans, cultural evolution has led to the accumulation and refinement of skills, tools and knowledge across generations. We provide a method for generating cultural transmission in artificially intelligent agents, in the form of few-shot imitation. Our agents succeed at real-time imitation of a human in novel contexts without using any pre-collected human data. We identify a surprisingly simple set of ingredients sufficient for generating cultural transmission and develop an evaluation methodology for rigorously assessing it. This paves the way for cultural evolution to play an algorithmic role in the development of artificial general intelligence.

Intelligence can be defined as the ability to acquire new knowledge, skills, and behaviours efficiently across a wide range of contexts[1]. Such knowledge represents the faculty to execute sequences of actions that achieve goals in appropriate contexts. Human intelligence is especially dependent on our ability to acquire knowledge efficiently from other humans. This knowledge is collectively known as culture, and the transfer of knowledge from one individual to another is known as *cultural transmission*. Cultural transmission is a form of social learning, learning assisted by contact with other agents. It is specialised for the acquisition of culture[2] via high fidelity, consistent recall, and generalisation to previously unseen situations. We refer to these properties as *robustness*[3]. Robust cultural transmission is ubiquitous in everyday human social interaction, particularly in novel contexts: copying a new recipe as seen on TV, following the leader on a guided tour, showing a colleague how the printer works, and so on.

We seek to generate an artificially intelligent agent capable of robust real-time cultural transmission from human co-players in a rich 3D physical simulation. The motivations for this are threefold. First, since cultural transmission is an ever-present feature of human social behaviour, it is a skill that an artificially intelligent agent should possess to facilitate beneficial human-AI interaction. Second, cultural

transmission can be seen as the process that underlies the evolution of culture[4], arguably the fastest known intelligence-generating process[5]. By exhibiting cultural transmission among embodied artificial agents in a complex space of 3D interactive tasks, we extend previous literature on computational models of cultural transmission[6] and cultural evolution[7] in the direction of using cultural evolution as an AI-generating algorithm[8,9]. Third, rich 3D physical simulations with sparse reward pose hard exploration problems for artificial agents, yet human behaviour in this setting is often highly informative and cheap in small quantities. Cultural transmission provides an efficient means of structured exploration in behaviour space.

We focus on a particular form of cultural transmission, known in the psychology and neuroscience literature as observational learning[10] or (few-shot) imitation[11]. In this field, imitation is defined to be the copying of body movement. It is a particularly impressive form of cultural transmission because it requires solving the challenging "correspondence problem"[12,13], instantaneously translating a sensory observation of another agent's motor behaviour into a motor reproduction of that behaviour oneself. In humans, imitation provides the kind of high-fidelity cultural transmission needed for the cultural evolution of tools and technology[14,15]. On the other hand, few-shot

---

[1]Google DeepMind, 6-8 Handyside Street, London N1C 4UZ, UK. ✉e-mail: edwardhughes@google.com

imitation is by no means the only form of cultural transmission[16]. Here, we provide a recipe for learning a few-shot imitation ability tabula rasa. Nevertheless, many of our methods are independent of the particular setting of imitation. We want to know how the skill of cultural transmission can develop when an artificial agent is learning from scratch, analogous in the cognitive science literature to the combination of phylogeny and ontogeny[17,18].

Our artificial agent is parameterised by a neural network and we use deep reinforcement learning (RL)[19] to train the weights. After training, the network is capable of few-shot, high-recall cultural transmission within a "test" episode, across a wide range of previously unseen tasks. Our approach contrasts with prior methods in which the training itself is a process of cultural transmission, namely imitation learning[20–22] and policy distillation[23,24]. These prior methods are highly effective on individual tasks, but they are not few-shot learners, require privileged access to datasets of human demonstrations or a target policy, and struggle to generalise to held-out tasks[25]. Our method amortises imitation learning into a real-time capability based on a single-stream of experience, which can be deployed on previously unseen tasks with no further training. This is important for many real-world applications, from construction sites to household robots, in which human data collection is costly, the tasks have inherent variation, and privacy is at a premium.

The central novelty in this work is the application of agent-environment co-adaptation[26,27] to generate an agent capable of robust real-time cultural transmission. To this end, we introduce a new open-ended reinforcement learning environment, *GoalCycle3D*. In this environment, the overall objective is to navigate between goal spheres in the correct order to accrue reward, avoiding obstacles along the way. The environment offers a rich diversity of task variants by virtue of procedural generation, 3D rigid-body physics, and continuous first-person sensorimotor control. Here, we explore more challenging exploration problems and generalisation challenges than in previous literature. We introduce a rigorous *cultural transmission metric* and transplant a two-option paradigm from cognitive science[28–30] to make causal inference about information transfer from one individual to another. This puts us on a firm footing from which to establish state-of-the-art generalisation and recall capabilities. Prior work[31–33] has used RL to generate test-time social learning, but these agents do not show within-episode recall or across-task generalisation.

Via careful ablations, we identify a minimal sufficient "starter kit" of training ingredients required for cultural transmission to emerge in GoalCycle3D, namely function approximation, memory (M), the presence of an expert co-player (E), expert dropout (D), attentional bias towards the expert (AL), and automatic domain randomisation (ADR). We refer to this collection by the acronym MEDAL-ADR. Memory is implemented as an LSTM network in the agent architecture. Our expert co-players are hard-coded bots, and are dropped in and out probabilistically during training episodes. This probabilistic dropout provides the right experience for agents to learn to observe what a useful demonstrator is doing and then remember and reproduce it when the demonstrator is absent. Attentional bias towards the expert is learned via an auxiliary loss to predict the position of the co-player. ADR gradually expands the distribution of tasks on which an agent trains, while maintaining a high cultural transmission capability. These components are ablated in turn in "The role of memory, expert demonstrations and attention loss" to "ADR for cultural transmission in complex worlds": only when all of them are acting in concert does robust cultural transmission arise in complex worlds.

Individually, these components aren't complex, but together they generate a powerful agent. We analyse the capabilities and limitations of our agent's cultural transmission abilities on three axes inspired by the cognitive science of imitation, namely recall, generalisation, and fidelity[34–37]. Recall assesses how well an agent can reproduce a demonstration without an expert present. Generalisation measures whether an agent can perform cultural transmission on held-out tasks. Fidelity computes to what extent an agent's choices closely match those of the expert demonstrator. We find that cultural transmission generalises outside the training distribution, and that agents recall demonstrations within a single episode long after the expert has departed. Introspecting the agent's "brain", we find strikingly interpretable neurons responsible for encoding social information and goal states.

## Results

### GoalCycle3D task space

We introduce GoalCycle3D, a 3D physical simulated task space built in Unity[38,39] which expands on the GoalCycle gridworld environment of ref. [33]. By anchoring our task dynamics to this previous literature and translating it to a 3D space, our results naturally extend prior work to a more naturalistic and realistic environment. The resulting richness is an important direction for the eventual deployment of AI, highlighting which algorithmic novelties are required to exceed the prior state-of-the-art in a more realistic setting.

Similar to ref. [27], we decompose an agent's task as the direct product of a world, a game and a set of co-players. The world comprises the size and topography of the terrain and the locations of objects. The game defines the reward dynamics for each player, which in GoalCycle3D amounts to a correct ordering of goals. A co-player is another interactive policy in the world, consuming observations and producing actions. Each task can be viewed as a different Markov decision process, thus presenting a distribution of environments for reinforcement learning.

While the 3D task space yields significant richness, it also presents opportunities for handcrafting which would reduce the generality of our findings. To avoid this, we make use of procedural generation over a wide task space. More specifically, we generate worlds and games uniformly at random for training, and test generalisation to held-out "probe tasks" at evaluation time, including a held-out human co-player, as described in "Probe Tasks". This train-test split provides data that enables overfitting to be ruled out, just as in supervised learning.

Worlds are parameterised by world size, terrain bumpiness and obstacle density. The obstacles and terrain create navigational and perception challenges for players. Players are positively rewarded for visiting goal spheres in particular cyclic orders. To construct a game, given a number of goals $n$, an order $\sigma \in S_n$ is sampled uniformly at random. The positively rewarding orders for the game are then fixed to be $\{\sigma, \sigma^{-1}\}$ where $\sigma^{-1}$ is the opposite direction of the order $\sigma$. An agent has a chance $\frac{2}{(n-1)!}$ of selecting a correct order at random at the start of each episode. In all our training and evaluation we use $n \geq 4$, so one is always more likely to guess incorrectly. The positions and orders of the goal spheres are randomly sampled at the start of each episode.

Players receive a reward of +1 for entering a goal in the correct order, given the previous goals entered. The first goal entered in an episode always confers a reward of +1. If a player enters an incorrect goal, they receive a reward of −1 and must now continue as if this were the first goal they had entered. If a player re-enters the last goal they left, they receive a reward of 0. The optimal policy is to divine a correct order, by experimentation or observation of an expert, and then visit the spheres in this cyclic order for the rest of the episode. Figure 1 summarises the GoalCycle3D task space.

### Measuring cultural transmission

The term cultural transmission has a variety of definitions, reflecting the diverse literature on the subject. For the purpose of clarity, we adopt a specific definition in this paper, one that captures the key features of few-shot imitation. Intuitively, the agent must improve its performance upon witnessing an expert demonstration and maintain that improvement within the same episode once the demonstrator has departed. However, what seems like test-time cultural transmission

might actually be cultural transmission during training, leading to memorisation of fixed navigation routes. To address this, we measure cultural transmission in held-out test tasks and with human expert demonstrators[40,41], similar to the familiar train-test dataset split in supervised learning[42].

Capturing this intuition, we define *cultural transmission* from expert to agent to be the average of improvement in agent score when an expert is present and improvement in agent score when that expert has subsequently departed, normalised by the expert score, evaluated on held-out tasks that have never before been experienced by the agent. Mathematically, let $E$ be the total score achieved by the expert in an episode of a held-out task. Let $A_{full}$ be the score of an agent with the expert present for the full episode. Let $A_{solo}$ be the score of the same agent without the expert. Finally, let $A_{half}$ be the score of the agent with the expert present from the start to halfway into the episode. Our metric of cultural transmission is

$$\text{CT} := \frac{1}{2}\frac{A_{full} - A_{solo}}{E} + \frac{1}{2}\frac{A_{half} - A_{solo}}{E}. \tag{1}$$

A completely independent agent doesn't use any information from the expert. Therefore it has a value of CT near 0. A fully expert-dependent agent has a value of CT near 0.75. An agent that follows perfectly when the expert is present, but continues to achieve high scores once the expert is absent has a value of CT near 1. This is the

desired behaviour of an agent from a cultural transmission perspective, since the knowledge about how to solve the task was transmitted to, retained by and reproduced by the agent.

## Cultural transmission is a bridge to adaptation

We first examine how reinforcement learning can generate cultural transmission in a relatively simple setting, a 4-goal game in a $20 \times 20\,\text{m}^2$ empty world. This is far from the most challenging task space for our algorithm, but it has a simplicity that is useful for developing our intuition. We find that an agent trained with memory (M), expert dropout (ED), and an attention loss (AL) on tasks sampled in this subspace experiences 4 distinct phases of training. The learning pathway of the agent passes through a cultural transmission phase to reach a policy that is capable of online adaptation, experimenting to discover and exploit the correct cycle within a single episode. By comparison, a vanilla RL baseline (M) is incapable of learning this few-shot adaptation behaviour. In fact it completely fails to get any score on the task (see "The role of memory, expert demonstrations and attention loss"). Cultural transmission, then, is functioning as a bridge to few-shot adaptation.

The training cultural transmission metric shows four distinct phases over the training run, each corresponding to a distinct social learning behaviour of the agent (see Fig. 2). In phase 1 (red), the agent starts to familiarise itself with the task, learns representations, locomotion, and explores, without much improvement in score. In phase 2

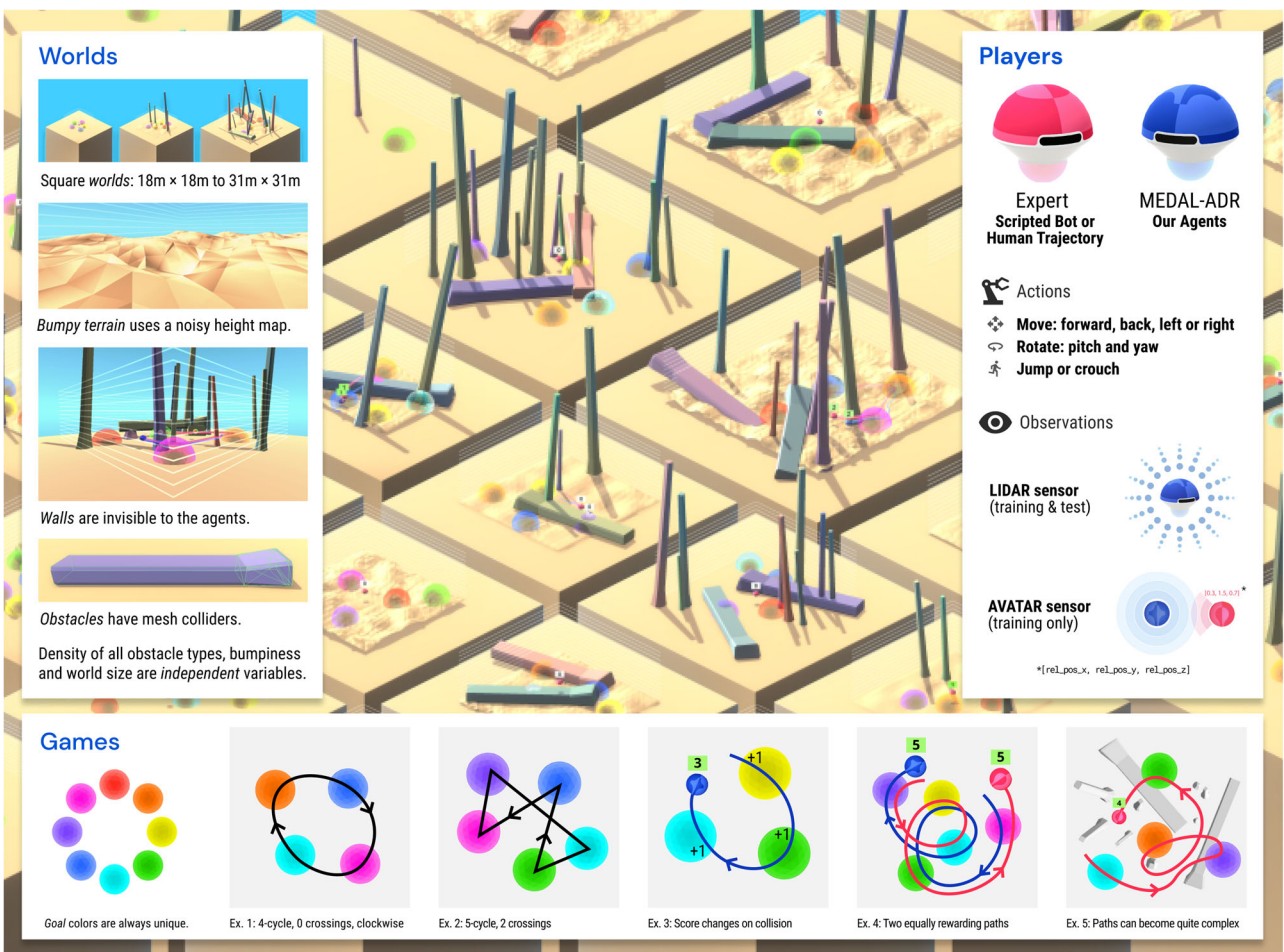

**Fig. 1 | GoalCycle3D.** A 3D physical simulated task space. Each task contains procedurally generated terrain, obstacles, and goal spheres, with parameters randomly sampled on task creation. Each agent is independently rewarded for visiting goals in a particular cyclic order, also randomly sampled on task creation. The correct order is not provided to the agent, so an agent must deduce the rewarding order either by

experimentation or via cultural transmission from an expert. Our task space presents navigational challenges of open-ended complexity, parameterised by world size, obstacle density, terrain bumpiness and a number of goals. Our agent observes the world using LIDAR (see Supplementary Movie 30).

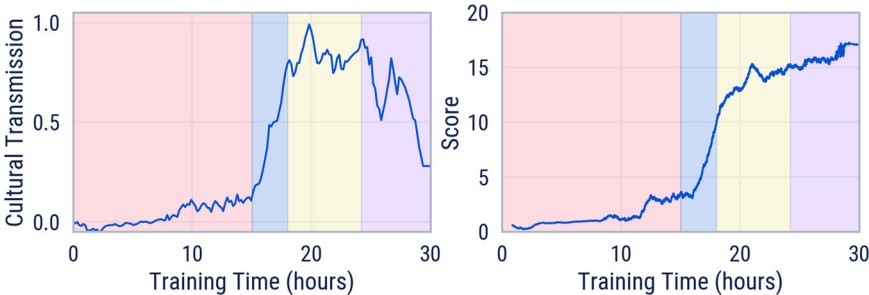

**Fig. 2 | Training without ADR.** Training cultural transmission (left) and agent score (right) for training without ADR on 4-goal in a small empty world. Colours indicate four distinct phases of agent behaviour from left to right: (1) (red) startup and exploration, (2) (blue) learning to follow, (3) (yellow) learning to remember, (4) (purple) becoming independent from expert.

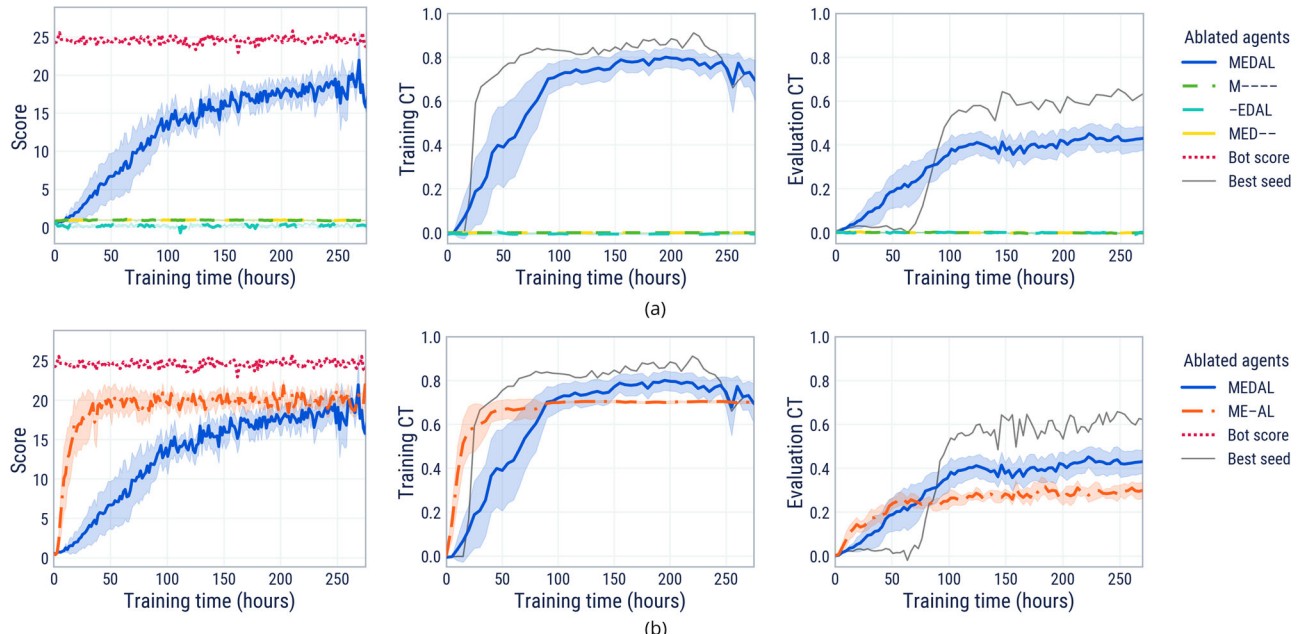

**Fig. 3 | Ablations of MEDAL ingredients.** Score (left), training cultural transmission (CT, centre), and evaluation CT on empty world 5-goal probe tasks (right) over the course of training. **a** Comparing MEDAL with three ablated agents, each trained without one crucial ingredient: without an expert (M——), memory (–EDAL), or attention loss (MED−). **b** Ablating the effect of expert dropout, comparing no dropout (ME−AL) with expert dropout (MEDAL). We report the mean performance for each across 10 initialisation seeds for agent parameters and task procedural generation. We also include the expert's score and MEDAL's best seed for scale and upper-bound comparisons. The shaded area on the graphs is one standard deviation.

(blue), with sufficient experience and representations shaped by the attention loss, the agent learns its first social learning skill of following the expert bot to solve the task. The training cultural transmission metric increases to 0.75, which suggests pure following.

In phase 3 (yellow), the agent learns the more advanced social learning skill that we call cultural transmission. It remembers the rewarding cycle while the expert bot is present and retrieves that information to continue to solve the task when the bot is absent. This is evident in a training cultural transmission metric approaching 1 and a continued increase in agent score.

Lastly, in phase 4 (purple), the agent is able to solve the task independent of the expert bot. This is indicated by the training cultural transmission metric falling back towards 0 while the score continues to increase. The agent has learned a memory-based policy that can achieve high scores with or without the bot present. More precisely, MEDAL displays an "experimentation" behaviour in this phase, which involves using hypothesis-testing to infer the correct cycle without reference to the bot, followed by exploiting that correct cycle more efficiently than the bot does (see Supplementary Movies 1–4). The bot

is not quite optimal because for ease of programming it is hard-coded to pass through the centre of each correct goal sphere, whereas reward can be accrued by simply touching the sphere. Note by comparison with Fig. 3a that this experimentation behaviour does not emerge in the absence of prior social learning abilities.

In other words, few-shot imitation creates the right prior for few-shot adaptation to emerge, which remarkably leads to improvement over the original demonstrator's policy. Note that, social learning by itself is not enough to generate experimentation automatically, further innovation by reinforcement learning, on top of the culturally transmitted prior, is necessary for the agent to exceed the capabilities of its expert partner. Our agent stands on the shoulders of giants, and then riffs to climb yet higher.

**The role of memory, expert demonstrations and attention loss**

We have shown that our MEDAL agent is capable of learning a test-time cultural transmission ability. Now, we show that the set of ingredients is minimal, by demonstrating the absence of cultural transmission when any one of them is removed. In every experiment, MEDAL and its

ablated cousins were trained on procedurally generated 5-goal, $20 \times 20$ worlds with no vertical obstacles and horizontal obstacles of density $0.0001\,m^{-2}$, and evaluated on the empty world 5-goal probes in "Probe tasks". We use a variety of different dropout schemes, depending on the ablation. M−- is trained with full dropout (expert is never present), ME−AL is trained with no dropout (expert is always present) and all other agents are trained with probabilistic dropout.

Figure 3a shows that memory (M), the presence of an expert (E), and our attention loss (AL) are important ingredients for the learning of cultural transmission. In the absence of these the agent achieves 0 score and therefore also doesn't pick up any reward-influencing social cues from the expert (if present), accounting for a mean CT of 0.

First, we consider the M−-- ablation. By removing expert demonstrations and, consequently, all dependent components, the dropout (D) and attention loss (AL), the agent must learn to determine the correct goal ordering by itself in every episode. The MPO agent's exploration strategy is not sufficiently structured to deduce the underlying conceptual structure of the task space, so the agent simply learns a "risk-averse" behaviour of avoiding goal spheres altogether (see Supplementary Movie 5).

Next, we analyse the −EDAL ablation. Without memory, our agent cannot form connections to previously seen cues, be they social, behavioural, or environmental. When replacing the LSTM with an equally sized MLP (keeping the same activation functions and biases, but removing any recurrent connections), our agent's ability to register and remember a solution is reduced to zero.

Lastly, we turn to the MED− ablation. Having an expert at hand is futile if the agent cannot recognise and pay attention to it. When we turn off the attention loss, the resulting agent treats other agents as noisy background information, attempting to learn as if it were alone. Vanilla reinforcement learning benefits from social cues to bootstrap knowledge about the task structure; the attention loss encourages it to recognise social cues. Note that the attention loss, like all auxiliary losses to shape neural representations, is only required at training time. This means that our agent can be deployed with no privileged sensory information at test time, relying solely on its LIDAR.

### Expert dropout enables within-episode memorisation

To isolate the importance of expert dropout, we compare our MEDAL agent (in which the expert intermittently drops in and out) with the previous state-of-the-art method ME-AL (in which the expert is always present). We use the same procedural generation and evaluation setting as in the previous section. Studying Fig. 3b, we see that the addition of expert dropout to the previous state of the art leads to better CT. MEDAL achieves higher CT both during training and when evaluated on empty world 5-goal probe tasks. This is because dropout encourages the learning of within-episode memorisation, a capability that was absent from previous agents[33] and which confers a higher cultural transmission score (see also "Agents recall expert demonstrations with high fidelity").

### ADR for cultural transmission in complex worlds

As we have seen, learning cultural transmission in a fixed task distribution acts as a gateway for learning few-shot adaptation. While this is undeniably useful in its own right, it begs the question: how can an agent learn to transmit cultural information in more complex tasks? ADR is a method of expanding the task distribution across training time to maintain it in the "Goldilocks zone" for cultural transmission. It gradually increases the complexity of the training worlds in an open-ended procedurally generated space (parameterised by 7 hyperparameters).

Figure 4a shows an example expansion of the randomisation ranges for all parameters for the duration of an experiment. Training CT is maintained between the boundary update thresholds 0.75 and 0.85. We see an initial start-up phase of ~100 hours when social

learning first emerges in a small, simple set of tasks. Once training CT exceeds 0.75, all randomisation ranges began to expand. Different parameters expand at different times, indicating when the agent has mastered different skills such as jumping over horizontal obstacles or navigating bumpy terrain. For intuition about the meaning of the parameter values, see Supplementary Movies 6–9.

To understand the importance of ADR for generating cultural transmission in complex worlds, we ablate the automatic (A) and domain randomisation (DR) components of MEDAL-ADR (for parameter values, see Supplementary Table D.1). The MEDAL agent is trained on worlds as complicated as the end point of the ADR curriculum. The MEDAL-DR agent is trained on a uniformly sampled distribution between the minimal and maximal complexities of the ADR curriculum (i.e., no automatic adaptation of the curriculum). In Fig. 4b we observe that ADR is crucial for the generation of cultural transmission in complex worlds, with MEDAL-ADR achieving significantly higher scores and cultural transmission than both MEDAL-DR and MEDAL.

### Agents recall expert demonstrations with high fidelity

To demonstrate the recall capabilities of our best-performing agent, we quantify its performance across a set of tasks where the expert drops out. The intuition here is that if our agent is able to recall information well, then its score will remain high for many timesteps even after the expert has dropped out. However, if the agent is simply following the expert or has poor recall, then its score will instead drop immediately close to zero. To our knowledge, within-episode recall of a third-person demonstration has not previously been shown to arise from reinforcement learning. This is an important discovery, since the recent history of AI research has demonstrated the increased flexibility and generality of learned behaviours over pre-programmed ones. What's more, third-person recall within an episode amortises imitation onto a timescale of seconds and does not require perspective matching between co-players. As such, we achieve the fast adaptation benefits of previous first-person few-shot imitation works (e.g., refs. 22,43,44) but as a general-purpose emergent property from third-person RL rather than via a special-purpose first-person imitation algorithm.

For each task, we evaluate the score of the agent across ten contiguous 900-step trials, comprising an episode of experience for the agent. In the first trial, the expert is present alongside the agent, and thus the agent can infer the optimal path from the expert. From the next trial onwards the expert is dropped out and therefore the agent must continue to solve the task alone. The world, agent, and game are not reset between trial boundaries; we use the term "trial" to refer to the bucketing of score accumulated by each player within the time window. We consider recall from two different experts, a scripted bot and a human player. For both, we use the worlds from the 4-goal probe tasks (see "Automatic domain randomisation").

Figure 5 compares the recall abilities of our agent trained with expert dropout (MEDAL-ADR) and without (ME-AL, similar to the prior state of the art[33]). Notably, after the expert has dropped out, we see that our MEDAL-ADR agent is able to continue solving the task for the first trial while the ablated ME-AL agent cannot. MEDAL-ADR maintains a good performance for several trials after the expert has dropped out, despite the fact that the agent only experienced 1800-step episodes during training. From this, we conclude that our agent exhibits strong within-episode recall.

**Evidence of causality.** To show causal information transfer from the expert to the agent in real time, we can adopt a standard method from the social learning literature. In the "two-action task"[28–30] subjects are required to solve a task with two alternative solutions. Half of the subjects observe a demonstration of one solution while the others observe a demonstration of the alternative solution. If subjects disproportionately use the observed solution, this is evidence that

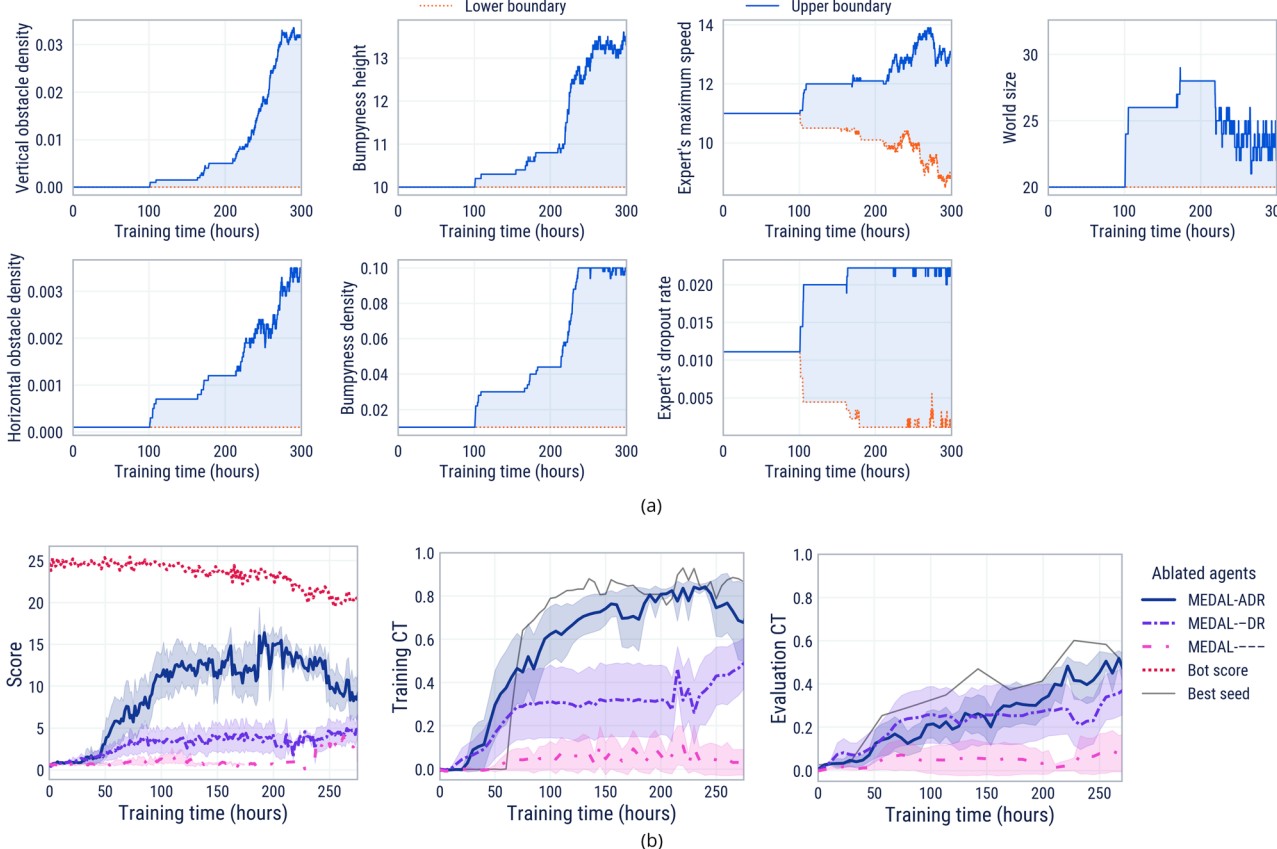

**Fig. 4 | Analysis of ADR parameter expansion and ablation of ADR ingredients. a** The expansion of parameter ranges over training for one representative seed in MEDAL-ADR training. **b** Score (left), training Cultural Transmission (CT, centre), and evaluation CT on complex world probe tasks (right) over the course of training for the automatic (A) and domain randomisation (DR) ablations of MEDAL-ADR. We report the mean performance for each across 10 initialisation seeds for agent parameters and task procedural generation. We also include the expert's score and the best MEDAL-ADR seed for scale and upper bound comparisons. The shaded area on the graphs is one standard deviation.

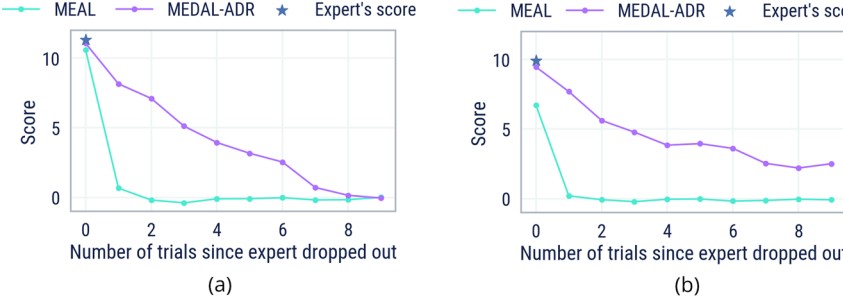

**Fig. 5 | Agent recall.** Score of MEDAL-ADR and ME-AL agents across trials since the expert dropped out. **a** Experts are scripted bots. **b** Experts are human trajectories. Supplementary Movie 10 shows MEDAL-ADR's recall from a bot demonstration in a 3600-step (4 trial) episode. Supplementary Movie 31 shows MEDAL-ADR's recall from a human demonstration in an 1800-step (2 trial) episode.

supports imitation. This experimental approach is widely used in the field of social learning; we use it here as a behavioural analysis tool for artificial agents for the first time. Using the tasks from our game space analysis, we record the preference of the agent in pairs of episodes where the expert demonstrates the optimal cycles $\sigma$ and $\sigma^{-1}$. The preference is computed as the percentage of correct complete cycles that an agent completes that match the direction of the expert cycle. Evaluating this over 1000 trials, we find that the agent's preference matched the demonstrated option 100% of the time, i.e., in every completed cycle of every one of the 1000 trials.

Trajectory plots further reveal the correlation between expert and agent behaviour (see Fig. 6). By comparing trajectories under different conditions, we can again argue that cultural transmission of information from expert to agent is causal. The agent cannot solve the task when the bot is not placed in the environment (Fig. 6a). When the bot is placed in the environment, the agent is able to successfully reach each goal and then continue executing the demonstrated trajectory after the bot drops out (Fig. 6b). However, if an incorrect trajectory is shown by the expert, the agent still continues to execute the wrong trajectory (Fig. 6c).

## Agents generalise across a wide task space

To demonstrate the generalisation capabilities of our agents, we quantify their performance over a distribution of procedurally

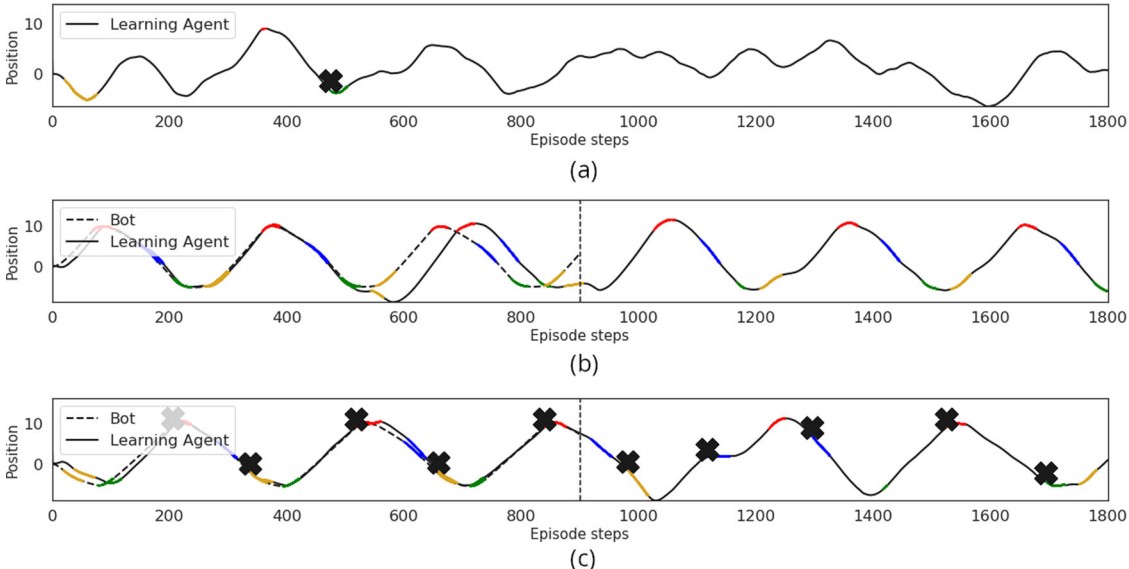

**Fig. 6 | Evidence of causality.** Trajectory plots for MEDAL-ADR agent for a single episode. **a** The bot is absent for the whole episode. **b** The bot shows a correct trajectory in the first half of the episode and then drops out. **c** The bot shows an incorrect trajectory in the first half of the episode and then drops out. The coloured parts of the lines correspond to the colour of the goal sphere the agent and expert have entered and the × s correspond to when the agent entered the incorrect goal. Here, position refers to the agent's position along the z-axis. Supplementary Movies 11–13 correspond to each plot respectively.

generated tasks, varying the underlying physical world and the over-lying goalcycle game. We analyse both "in-distribution" and "out-of-distribution" generalisation, with respect to the distribution of parameters seen in training (see Supplementary Table C.2). Out-of-distribution values are calculated as ±20% of the min/max in-distribution ADR values where possible, and indicated by cross-hatched bars in all figures.

In every task, an expert bot is present for the first 900 steps, and is dropped out for the remaining 900 steps. We define the *normalised score* as the agent's score in 1800 steps divided by the expert's score in 900 steps. An agent who can perfectly follow but cannot remember will score 1. An agent which can perfectly follow and can perfectly remember will score 2. Values in between correspond to increasing levels of cultural transmission.

**World space.** The space of worlds is parameterised by the size and bumpiness of the terrain (terrain complexity) and the density of obstacles (obstacle complexity). To quantify generalisation over each parameter in this space, we generate tasks with worlds sampled uniformly from the chosen parameter while setting the other parameters at their lowest in-distribution value. Games are then uniformly sampled across the possible number of crossings for 5 goals.

From Fig. 7a, we conclude that MEDAL-ADR generalises well across the space of worlds, demonstrating both following and remembering across the majority of the parameter variations considered, including when the world is out-of-distribution.

**Game space.** The space of games is defined by the number of goals in the world as well as the number of crossings contained in the correct navigation path between them. To quantify generalisation over this space, we generate tasks across the range of feasible "*N*-goal *M*-crossing" games in a flat empty world.

Figure 7b shows our agent's ability to generalise across games, including those outside of its training distribution. Notably, MEDAL-ADR can perfectly remember all numbers of crossings for the in-distribution 5-goal game. We also see impressive out-of-distribution generalisation, with our agent exhibiting strong remembering, both in 4-goal and 0-crossing 6-goal games. Even in complex 6-goal games with many crossings, our agent can still perfectly follow.

## Introspecting the agent's brain

Deep learning models are not necessarily readily interpretable. On the other hand, interpretability is often desirable or even pre-requisite for deploying AI systems in the real world. Here, we demonstrate that our model is interpretable at the neural level. Training agents to imitate via meta-reinforcement learning embeds the logic for a state-machine capable of approximately Bayes-optimal cultural transmission into the neural network's weights[45]. By inspecting a trained agent's memory, we find clearly interpretable individual neurons. These neurons have specialised roles required for solving a new task online via cultural transmission, a subset of the sufficient statistics which drive the state-machine[46]. One, dubbed the *social neuron*, encapsulates the notion of agency; the other, called the *goal neuron*, captures the periodicity of the task.

To identify the social neuron, we use linear probing[47,48], a well-known and powerful method for understanding intermediate layers of a deep neural network. We train an attention-based classifier to classify the presence or absence of an expert co-player based on the memory state of the agent. The neuron with the maximum attention weight is defined to be a social neuron, and its activation crisply encodes the presence or absence of the expert in the world (Fig. 8a). Figure 8b shows a stark difference in prediction accuracy for expert presence between differently ablated agents. This suggests that the attention loss (AL) is at least partly responsible for incentivising the construction of "socially-aware" representations.

To identify the goal neuron we inspect the variance of memory neural activations across an episode, finding a neuron whose activation is highly correlated with the entry of an agent into a goal sphere. Figure 8c shows that this neuron fires when the agent enters and remains within a goal sphere. Interestingly, it is not the presence or the following of an expert that determines the spikes, nor the observation of a positive reward. Appendix D.3 contains full details of our methods and results.

## Discussion

In this work, we have viewed robust cultural transmission as a "cognitive gadget"[2]. That is to say, it may be learned purely from individual reward[49], requiring only that the environment contains knowledgeable co-players, and that the agent has a minimal sufficient number of

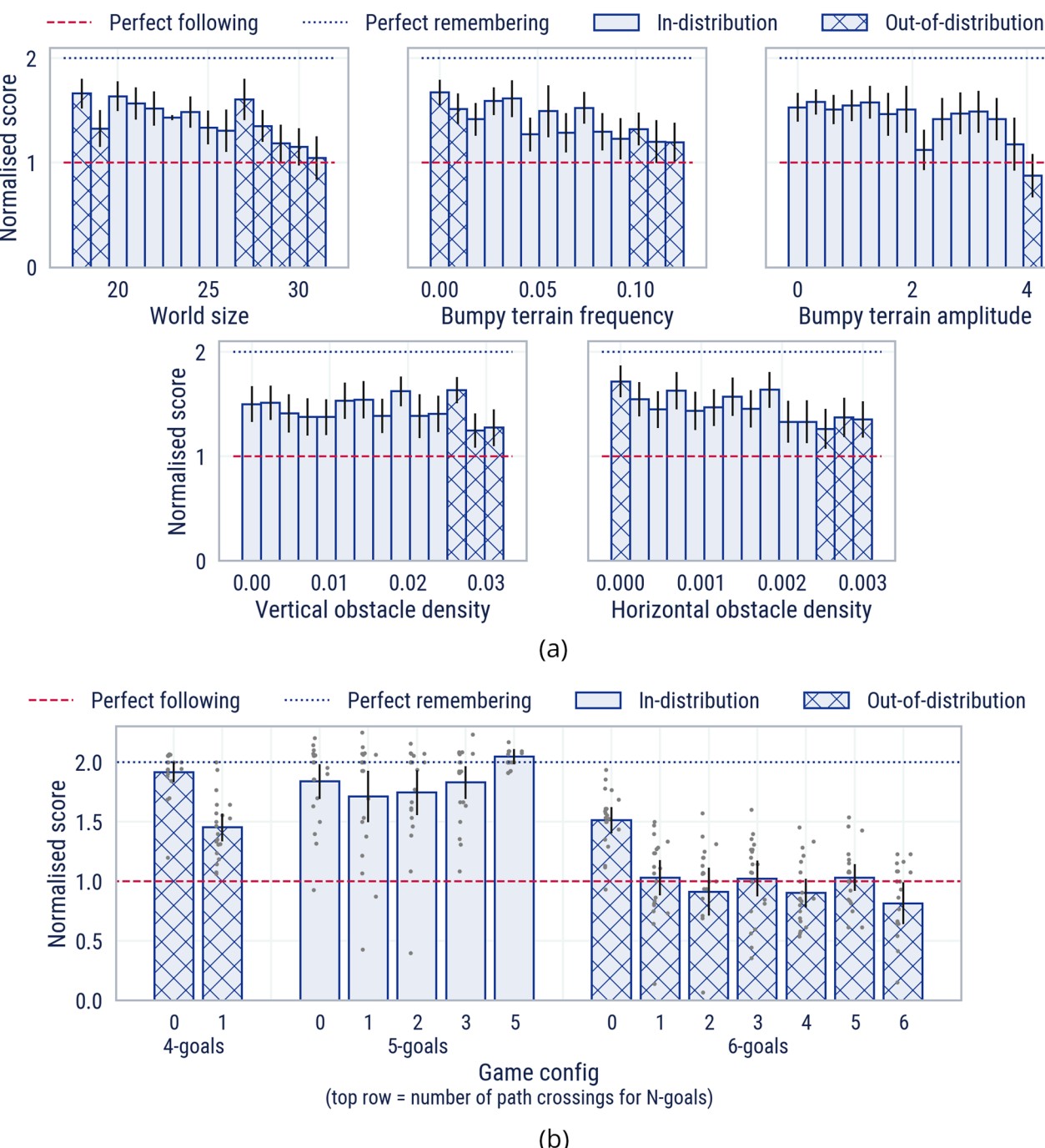

**Fig. 7 | Task space generalisation. a** A slice through the world space allows us to disentangle MEDAL-ADR's generalisation capability across different world space parameters. **b** MEDAL-ADR generalises across the game space, demonstrating remembering capability both inside and outside the training distribution. We report the mean performance across 50 initialisation seeds for **a** and 20 initialisation seeds for **b**. The error bars on the graphs represent 95% confidence intervals. Supplementary Movies 14–20 demonstrate generalisation over the world space and game space.

representational and experiential biases. All of these ingredients have analogues in human and animal cognition. Better working memory (M) is known to correlate with improved fluid intelligence in humans[50] and the ability to solve novel reasoning problems[51], including by imitation[52]. The progressive increase in duration of expert dropout (ED) mirrors the development of secure attachment in humans[53], and is important for the learning of imitation in animals[54]. Humans, along with many animals, have an in-built attentional bias towards biological motion[55,56], mirroring our attention loss (AL). Among animals, social learning is known to be preferred over asocial learning in uncertain or

varying ecological contexts[57,58], environment properties we create via ADR.

We can characterise our approach to generating cultural transmission as memory-based meta-learning[59,60]. After training, our agent can infer the expert's policy online using only a single stream of experience. This approach has several benefits for real-world deployment. Imagine deploying a robot in a kitchen. One would hope that the robot would adapt quickly online if the spoons are moved, or if a new chef turns up with different skills. Moreover, one might have privacy concerns about relaying large quantities of data to a central server for

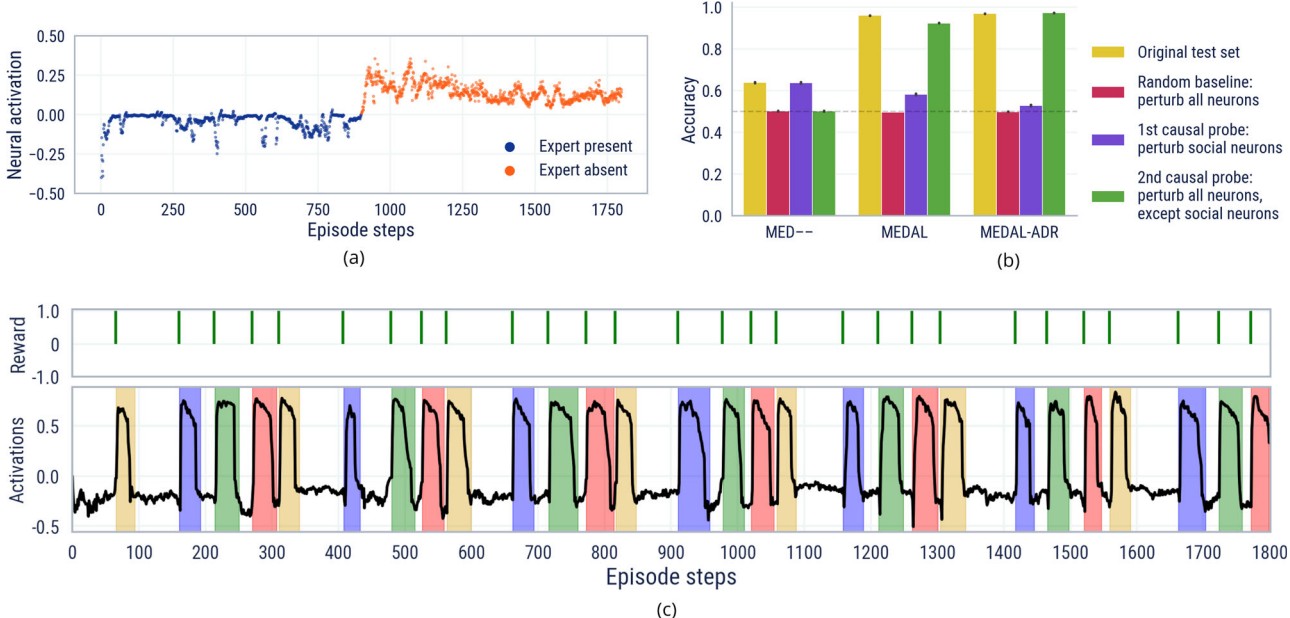

**Fig. 8 | Introspection of Agent's Brain. a** Activations for MEDAL-ADR's social neuron. **b** We report the accuracy of three linear probing models trained to predict the expert's presence based on the belief states of three agents (MED−, MEDAL, and MEDAL-ADR). We make two causal interventions (in green and purple) and a control check (in red) on the original test set (yellow). We report the mean performance across 10 different initialisation seeds. The small standard deviation error bars suggest a broad consensus across the 10 runs on which neurons encode social information. **c** Spikes in the goal neuron's activations correlate with the time the agent remains inside a goal (illustrated by coloured shading). The goal neuron was identified using a variance analysis, rather than the linear probing method in **b**.

training. Our agent adapts to human cultural data on-the-fly and within its local memory, and thus is both robust and privacy-preserving.

From a safety perspective, our trained agent as an artefact has a case of objective misgeneralisation[61], which we see in Fig. 6c. The agent happily follows an incorrect demonstration and even reproduces that incorrect path once the expert has dropped out. This is unsurprising since the agent has only encountered near-perfect experts in training. To address this, an agent could be trained with a variety of experts, including noisy experts and experts with objectives that are different from the agent's. With such experience, an agent should meta-learn a selective social learning strategy[62,63], deciding what knowledge to infer from which experts, and when to rely on independent exploration.

We identify three limitations with our evaluation approach. Firstly, we did not test cultural transmission from a range of humans, but rather with a single player from within the study team. Therefore we cannot make statistically significant claims about robustness across a human population. Secondly, the diversity of reasonable human behaviour is constrained by our navigation task. To gain more insight into generalisable cultural transmission, we need tasks with greater strategic breadth and depth. Lastly, we do not distinguish whether our trained agents memorise a geographical path, or whether they memorise an abstraction over the correct sphere ordering. To disambiguate this, one could change the geographical location of goal spheres at the moment of expert dropout but leave the ordering the same.

It is natural to ask whether our MEDAL-ADR recipe is sufficient more generally, outside the GoalCycle3D task space. The answer is a qualified "no". In our favour, GoalCycle3D is already a large, procedurally generated task space. Moreover it can be seen as the navigational representative for an even bigger class of tasks: those which require a repeated sequence of strategic choices, such as cooking, wayfinding, and problem solving. It is reasonable to expect that similar methods would work well in other representative environments from this class of tasks. However, there are environmental affordances that we necessarily assume for our method, including expert visibility, dropout and procedural generation. If these are impossible to create or approximate in an environment, then our method cannot be applied. More subtly there are silent assumptions: that finding an initial reward is relatively easy, there is no fine-motor control necessary, the timescale for an episode is relatively short, there are no irreversibilities, the goals are all visible, the rewarding order remains constant throughout an episode. We hope that future studies will relax each of these requirements, creating new challenges for research.

Is a more minimal "starter kit" possible in a different task space? Let us first assume that we remain within the framework of deep RL. We know that the MEAL variant[33] does not learn a within-episode recall ability, even in a gridworld, so expert dropout is likely to be essential. Clearly, the memory component is necessary for successful meta-learning, which leaves attention loss and ADR. The necessity of attention loss is a function of the relevant salience of the expert with respect to the rest of the environment. In a sufficiently simple environment, or with a sufficiently powerful learning algorithm, this loss might become superfluous. However, drawing an analogy with humans suggests that an attention bias might be an important and general built-in shortcut to social learning: infants display such a bias from a very young age[64]. As we have seen, ADR is not a pre-requisite for cultural transmission to emerge, but rather a technique for maintaining this behaviour during open-ended learning. As machine learning algorithms become increasingly powerful, we expect that such unsupervised environment design algorithms[65] will become a ubiquitous tool for avoiding overfitting and maintaining within-episode adaptation. Of course, if we remove the assumption of the deep RL framework, it is quite possible that entirely different "starter kits" for cultural transmission exist: our work may stand as a lighthouse on a much longer research journey.

There are a wealth of natural extensions to this work, towards the goal of generating open-ended cultural evolution. Firstly, it would be interesting to bootstrap agent cultural capabilities from scratch using generational training[66], as opposed to relying on hand-coded expert bots as co-players. Distillation is already known to create a ratchet effect across generations[27], and cultural transmission can be viewed as amortised distillation, so we would expect this approach to generate efficient open-ended learning. Secondly, humans imitate over abstract

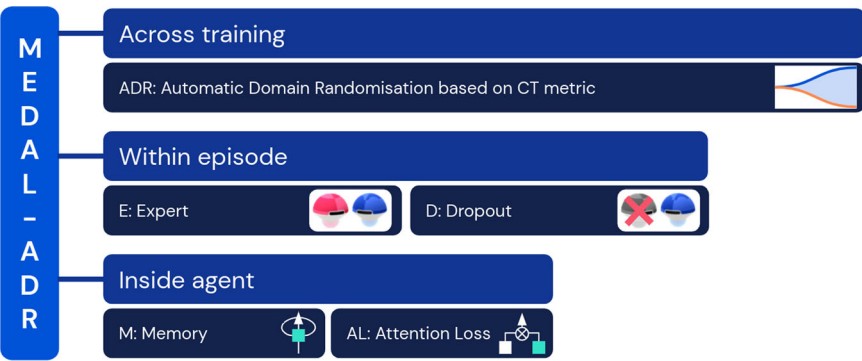

**Fig. 9 | Ingredients of MEDAL-ADR.** The minimal sufficient ingredients that comprise our methods, grouped by the timescale on which they operate.

representations of the task at hand, including beliefs, desires and intentions. Whether co-adaptation of training experience can lead to such "theory of mind" in artificial agents remains an open question. This approach would complement explicit model-based methods adopted in prior work (e.g., refs. 67,68). Finally, cultural transmission is a necessary condition for generating cultural evolution but may or may not be sufficient (see ref. 69 for a discussion). Earlier in the discussion, we argued that appropriate randomisation over experts may generate selective social learning. One might also ask for variation in behaviour space to power the evolutionary system. Fortunately, there are a variety of off-the-shelf techniques for generating diverse policies[70,71]. Bringing all of these components together, it would be fascinating to validate or falsify the hypothesis that cultural evolution in a population of agents may lead to the accumulation of behaviours that solve an ever-wider array of human-relevant real-world problems.

We do not view our MEDAL-ADR method as a direct model for the development of cultural transmission in humans. However, the time is ripe for such research. The experimental (e.g., refs. 72,73) and theoretical (e.g., refs. 2,74) fields are already well-developed, and this work provides plausible AI modelling techniques. Neuroscience and deep RL have already mutually benefited from collaborations with a modelling flavour[75], and the precedent has been set for MARL as a modelling tool in social cognition[76,77]. One could imagine experiments comparing the various ablated models of MEDAL-ADR with the behaviour of children at different stages of ontogeny, or with the behaviour of non-human animals, were one to design appropriate tasks that control for the various sensorimotor differences between AIs, humans and animals. Alternatively, one might consider a line of experiments that investigated cultural accumulation across several "generations" of humans and AIs in a laboratory environment, drawing comparisons between the different populations, or analysing the effects of mixing human and AI participants in a population. We look forward to fruitful interdisciplinary interaction between the fields of AI and cultural evolutionary psychology in the future.

## Methods

In our work, cultural transmission emerges from reinforcement learning augmented with a minimal sufficient set of ingredients, measured on held-out probe tasks. Together the ingredients are referred to as MEDAL-ADR, and are summarised in Fig. 9. The components modulate reinforcement learning on three distinct timescales. Within the agent's neural architecture, a memory module (M) builds a belief-state representation of the task frame-by-frame. During training, an attention loss (AL) shapes the neural representation towards paying attention to co-players. This loss is not required at test time. Zooming out to the timescale of an episode, the experience stream of the agent contains an expert co-player (E) which drops in and out (D) probabilistically. Finally, across training, the distribution of tasks experienced in different episodes by the agent is non-stationary. ADR creates a

curriculum of tasks designed to promote and maintain cultural transmission over an ever-wider task space. Our probe tasks can be thought of as a held-out "test set" for evaluating the generalisation capabilities of agents trained with RL. We now discuss each methodological contribution in detail.

### Reinforcement learning

We train our agent via distributed deep reinforcement learning (RL) using a state-of-the-art learning algorithm: maximum a posteriori policy optimisation (MPO)[78]. MPO is an actor-critic, model-free, continuous action-space deep reinforcement learning algorithm. Instead of using gradients from the $Q$-function, it leverages samples to compare different actions in a particular state, updating the policy to ensure that better (according to the current $Q$-function) actions have a higher probability of being sampled. As with other actor-critic algorithms, MPO alternates between *policy improvement* which updates the policy ($\pi$) using a fixed $Q$-function and *policy evaluation* which updates the estimate of the $Q$-function.

If our agent learns a cultural transmission policy, it is only by observing an expert agent in the world and correlating improved expected return with the ability to reproduce the other agent's behaviour. If that cultural transmission policy is robust, then RL must have favoured imitation with high fidelity, which generalises across a range of physical contexts, and where transmitted behaviours are recalled after the demonstrator has departed. For further details of the RL formalism and MPO algorithm, see Appendices C.1 and C.2. Our agent is trained using a large-scale distributed training framework, described in Appendix C.3. Details of the hyperparameters used for learning are reported in Appendix C.4.

### Memory (M)

The encoded observation is fed to a single-layer recurrent neural network (RNN) with an LSTM core[79] of size 512. This RNN is unrolled for 800 steps during training. The output of the LSTM, which we refer to as the *belief*, is passed to a policy, value and auxiliary prediction head. The policy and value heads together implement the MPO algorithm, while the prediction head implements the attention loss described in "Attention Loss (AL)". The AVATAR sensor observation is used as a prediction target for this loss.

### Expert dropout (ED)

Cultural transmission requires the acquisition of new behaviours from others. For an agent to robustly demonstrate cultural transmission, it is not sufficient to imitate instantaneously; the agent must also internalise this information, and later recall it in order to display the transmitted behaviour. We introduce *expert dropout* as a mechanism both to test for and to train for this recall ability.

At each timestep in an episode, the expert is rendered visible or hidden from the agent. Given a difficult exploration task that the agent

cannot solve through solo exploration, we can measure its ability to recall information gathered through cultural transmission by observing whether or not it is able to solve the task during the contiguous steps in which the expert is hidden. During training, we apply expert dropout in an automatic curriculum to encourage this recall ability, as described in "Automatic domain randomisation".

Mathematically, we formulate expert dropout as follows. Let $e_t \in \mathbb{Z}_2$ be the *state* of expert dropout at timestep $t$. State 0 corresponds to the expert being hidden at time $t$, by which we mean it will not be detected in any player's observation. State 1 corresponds to the expert being visible at time $t$. An expert dropout scheme is characterised by the state transition functions $e_{t+1} = f(e_t, t)$. We define the following schemes:

**No dropout.** $f(e_t, t) = 1$ for all $t$.

**Full dropout.** $f(e_t, t) = 0$ for all $t$.

**Half dropout.** For episodes of length $N$ timesteps,

$$f(e_t, t) = \begin{cases} 1 & t \le \lfloor N/2 \rfloor \\ 0 & t > \lfloor N/2 \rfloor \end{cases}. \tag{2}$$

**Probabilistic dropout.** Given transition probability $p \in [0, 1]$,

$$f(e_t, t) = \begin{cases} e_t + 1 \bmod 2 & \text{with probability } p \\ e_t & \text{with probability } 1 - p \end{cases}. \tag{3}$$

### Attention loss (AL)

To use social information, agents need to notice that there are other players in the world that have similar abilities and intentions as themselves[33,68]. Agents observe the environment without receiving other players' explicit actions or observations, which we view as privileged information. Therefore, we propose an *attention loss* that encourages the agent's belief to represent information about the current relative position of other players in the world. We use "attention" here in the biological sense, identifying what is important, in particular, that agents should pay attention to their co-players. Similar to previous work (e.g., ref. 80), we use a privileged AVATAR sensor as a prediction target, but not as an input to the neural network, so it is not required at test time.

Starting from the belief, we concatenate the agent's current action, pass this through two MLP layers of size 32 and 64 with *relu* activations, and finally predict the egocentric relative position of other players in the world at the current timestep. The objective is to minimise the $\ell^1$ distance between the ground truth and predicted relative positions. The attention loss is set to zero when the agent is alone (for instance, when the expert has dropped out).

### Automatic domain randomisation

An important ingredient for the development of cultural transmission in agents is the ability to train over a diverse set of tasks. Without diversity, an agent can simply learn to memorise an optimal route. It will pay no attention to an expert at test-time and its behaviour will not transfer to distinct, held-out tasks. This diverse set of tasks must be adapted according to the current ability of the agent. In humans, this corresponds to Vygotsky's concept of a dynamic Zone of Proximal Development (ZPD)[81]. This is defined to be the difference between a child's "actual development level as determined by independent problem solving" and "potential development as determined through problem solving under adult guidance or in collaboration with more capable peers". We also refer to the ZPD by the more colloquial term "Goldilocks zone", one where the difficulty of the task is not too easy nor too hard, but just right for the agent.

We use ADR[82] to maintain task diversity in the Goldilocks zone for learning a test-time cultural transmission ability. To apply ADR, each task must be parameterised by a set of $d$ parameters, denoted by $\lambda \in \mathbb{R}^d$. In *GoalCycle3D*, these parameters may be related to the world, such as terrain size or tree density, the game, such as number of goals, or the co-players, such as bot speed.

Each set of task parameters $\lambda$ are drawn from a distribution $P_\phi(\Lambda)$ over the $(d-1)$-dimensional standard simplex, parameterised by a vector $\phi$. We use a product of uniform distributions with $2d$ parameters and joint cumulative density function

$$P_\phi(\lambda) = \prod_{i=1}^{d} \frac{1}{\phi_i^H - \phi_i^L}, \tag{4}$$

defined over the standard simplex given by

$$\left\{ \lambda : \lambda_i \in \left[\phi_i^L, \phi_i^H\right] \text{ for } i \in \{1, \dots, d\}, \lambda \in \mathbb{R}^d \right\}. \tag{5}$$

Roughly speaking, the simplex boundaries $\phi_i^L$ or $\phi_i^H$ are expanded if the training cultural transmission metric exceeds an upper threshold

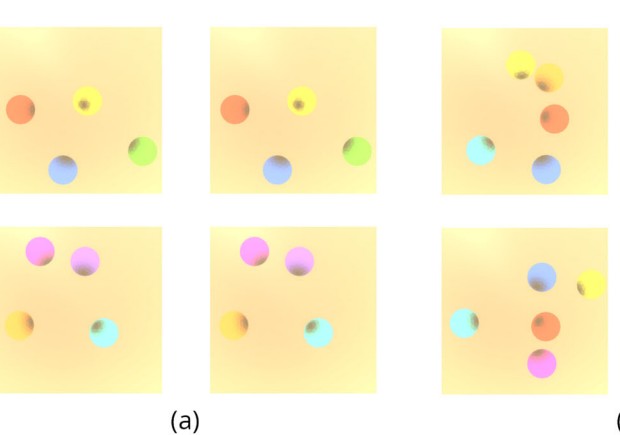
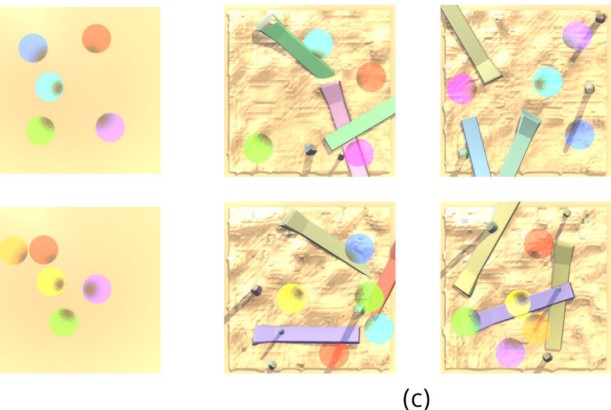

(a) (b) (c)

**Fig. 10 | Worlds and games used as probe tasks. a** Empty world, 4-goal games. **b** Empty world, 5-goal games. **c** Complex world, 4/5-goal games. These cover a representative range of crossings and colour combinations. The empty world probe tasks have terrain of size $20 \times 20 \, \text{m}^2$, while the complex world probe tasks have terrain of size $32 \times 32 \, \text{m}^2$. The complex world probes require clear examples of jumping behaviours and navigation around vertical obstacles. The human movement pattern in all probes is always goal-directed and near-optimal, but clearly different from a scripted bot, taking some time to get situated in the first few seconds and not taking an identical path on repeated cycles, for instance. See Supplementary Movies 21–29.

and contracted if the training cultural transmission metric drops below a lower threshold. This maintains the task distribution in the Goldilocks zone for learning cultural transmission. For more details, see Appendix C.5.

### Probe tasks

To better understand and compare the performance of our agents under specific conditions, we test them throughout training on a set of "probe tasks". These tasks are hand picked and held out from the training distribution (i.e., they are not used to update any neural network weights). The worlds and games used in our probe tasks are shown in Fig. 10.

Importantly, these tasks are not chosen based on agent performance. Instead, they are chosen to represent a wide space of possible worlds, games, and co-players. For example, we desire cultural transmission both in worlds devoid of any obstacles and in worlds that are densely covered. Consequently, we included both in our set of probe tasks. We save checkpoints of agents throughout training at regular intervals and evaluate each checkpoint on the probe tasks. This yields a held-out measure of cultural transmission at different points during training, and is a consistent measure to compare across independent training runs.

While we seek to generate agents capable of robust real-time cultural transmission from human co-players, it is infeasible to measure this during training at the scale necessary for conducting effective research. Therefore we create "human proxy" expert co-players for use in our probe tasks as follows. A member of the team plays each task alone and with privileged information about the optimal cycle. For each task, we record the trajectory taken by the human as they demonstrate the correct path. We then replay this trajectory to generate an expert co-player in probe tasks, disabling the human proxy's collision mesh to ensure that the human trajectory cannot be interfered with by the agent under test.

### Reporting summary

Further information on research design is available in the Nature Portfolio Reporting Summary linked to this article.

## Data availability

Data for this study was generated via a Unity-based simulation[38,39], with no additional external data sources. Source data for main text figures are provided with this paper, excluding Figs. 6 and 8c, for which analysis data was generated on-the-fly and not logged. Supplementary movies are provided with this paper. Source data are provided with this paper.

## Code availability

We are unable to release the code for this work as it was developed in a proprietary context. We are happy to answer specific questions concerning re-implementation: please contact reverett@deepmind.com. An open-source implementation of the MPO algorithm is available at https://github.com/deepmind/acme/tree/master/acme/agents/jax/mpo. An open-source implementation of the 2D variant of GoalCycle is available at https://github.com/kandouss/marlgrid/blob/master/marlgrid/envs/goalcycle.py. For data analysis, we used the following freely available packages: `numpy v1.25.2`, `pandas v1.5.3`, `matplotlib v3.6.1`, `seaborn v0.12.2`, `scipy v1.9.3`.

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

## Acknowledgements

We would like to thank many, both at DeepMind and in the wider community, for the conversations that have helped to shape this manuscript. We are particularly grateful to Lucy Aplin, Michael Azzam, Jakob Bauer, Satinder Baveja, Charles Blundell, Andrew Bolt, Kalesha Bullard, Max Cant, Nando de Freitas, Seb Flennerhag, Antonio Garcia Castañeda, Thore Graepel, Abhinav Gupta, Nik Hemmings, Max Jaderberg, Natasha Jaques, Andrei Kashin, Simon Kirby, Tom McGrath, Hamza Merzic, Alexandre Moufarek, Kamal Ndousse, Sherjil Ozair, Patrick Pilarski, Drew Purves, Thom Scott-Phillips, Ari Strandburg-Peshkin, DJ Strouse, Kate Tolstaya, Karl Tuyls, Marcus Wainwright, and Jane Wang. This work was funded entirely by DeepMind.

## Author contributions

A.B. contributed to learning process development, research investigations, infrastructure development, technical management, and manuscript editing. B.B. contributed to quality assurance testing. A.C. contributed to environment development. A.D. contributed agent analysis, infrastructure development, and code quality. A.E. contributed to research investigations and agent analysis. R.E. contributed cultural general intelligence concept, agent analysis, learning process development, research investigations, infrastructure development, additional environment design, technical management and team management. A.F. contributed to infrastructure development. Y.G. contributed to environment development. E.H. contributed cultural general intelligence concepts, research investigations, infrastructure development, team management, and manuscript editing. K.M. contributed to learning process development, research investigations, agent analysis, and team management. P.M. contributed to quality assurance testing. J.P. contributed to programme management. M.P. contributed to learning process development, research investigations, agent analysis, and infrastructure development. A.P. contributed environment visuals. E.S. contributed infrastructure development, technical management, and team management. S.S. contributed to programme management. A.Z. contributed to environment design, environment development, and agent analysis. L.Z. contributed to learning process development, research investigations, infrastructure development, and technical management.

## Competing interests

A patent has been registered covering aspects of this work. Details of the patent are as follows. Patent applicant: DeepMind Technologies Limited. Name of inventors: Bhoopchand, Avishkar Ajay; Collister, Adrian Ashley; Edwards, Ashley Deloris; Everett, Richard; Hughes, Edward Fauchon; Mathewson, Kory Wallace; Pîslar, Miruna; Zacherl, Alexander; Zhang, Lei. Application number: PCT/EP2023/055474. Status of application: pending. The specific aspect of manuscript covered in patent application: the specific method for generating cultural transmission represented in Fig. 9, instantiated as described in the Methods section and the Supplementary Information. The authors declare no other competing interests.
