## [Peer Review File · Nature Communications]

REVIEWER COMMENTS

Reviewer #1 (Remarks to the Author):

This paper presents a machine learning study of cultural transmission, introducing an agent capable of sophisticated imitation, memory, and generalization. Through careful comparison with ablated agents, the authors establish the minimal conditions for effective cultural transmission in their space of agents. Finally, they peek inside the "brain" of their agent to identify neurons that code for significant social and goal variables.

Overall, I thought this was a very nice paper. It is clearly written, accessible to non-specialists, thorough, rigorous, and appealing to a broad range of readers (including cognitive scientists). In my view the paper could be accepted more or less as is. I also want to point out that I am not an expert in this area, so my evaluation should be considered with that in mind.

I have a few minor comments:

- I'm not really sure what is being learned from the "introspection" analysis (section 2.9). The authors find some neurons that code for particular things, but it wasn't clear to me what that was telling us (obviously the same criticisms apply to neuroscientists doing similar things!).
- Please make sure that the error bars are identified in the figure captions. Also, sometimes standard deviation is reported, although standard error or 95% confidence intervals are more standard.
- This is probably outside the scope of the paper, but as a cognitive scientist I would have been interested in the extent to which this approach could be used to model what we know about cultural transmission in humans. Furthermore, it would be interesting to investigate whether the ablated versions of the model correspond to the more limited forms of cultural transmission observed in non-human animals. I think this might be worth at least a discussion point.

Reviewer #2 (Remarks to the Author):

This paper proposes a method, called MEDAL-ADR, to enable artificial intelligence (AI) agents to achieve few-shot imitation. Humans are good at observing/following an expert performing a task for some short time and then imitating them to achieve the task even when the expert is not present anymore. The goal of this paper is to equip the AI agents with this ability as well.

The proposed solution relies on reinforcement learning (RL), but the actual contribution is not the RL algorithm. Instead, the contribution of the paper is the investigation of components that are useful for training e.g., the expert should sometimes disappear to discourage the agent from simply copying the expert. The paper is well-written in general, and extensive experiments have been presented.

Below are my comments to the authors.

1- In the paper, the authors acknowledge this work is only about imitation whereas cultural transmission can be in many other forms. Given this, I think it is a little overselling to repeatedly mention and frame the work as achieving cultural transmission. The term "few-shot imitation", which is what this paper is really about, is already descriptive and interesting to the community.

2- Another point that I think is overselling is that the paper states they identify the sufficient components of the training to achieve few-shot imitation, and it also mentions some of the components as necessary based on experiment results. These statements carry mathematical claims, but there is no evidence. First, I disagree they are sufficient. The accurate statement would be they are sufficient for this specific environment. Because there is currently no evidence that MEDAL-ADR can achieve few-shot imitation in other environments as well, despite the claims of the current environment being very general. I also disagree the components are necessary. It might be possible that some other components (and their combinations) substitute the components of MEDAL-ADR that seems necessary in the experiments of this paper.

3- Why is the first term necessary / important in Equation 1? I think the second term is very well-motivated: the agent could learn simply copying the expert, so it is important to include a measure of success after the expert disappears. But it seems this second term already includes the cases where expert is present and absent. What does the first term contribute? Without the first term, a copycat agent (fully expert-dependent) would get a score of 0.5, which seems like a better scale.

4- Why is this score decreasing in Figure 2? Is it because the environment is so easy that the agent is able to solve it on its own and the existence of the expert does not contribute much? In other words, is it easier / more efficient for the agent to explore on its own rather than waiting for the expert performing the task? Does this mean the expert is not playing the game perfectly and it is possible to outperform it with RL? If yes, why does the paper make this choice (of simple environment and imperfect expert)? Please clarify.

5- Discussion says "an agent could be trained with a variety of experts, including noisy experts and experts with objectives that are different from the agent's." This would only make sense if the reward signal is part of the observation that the policy depends on, right?

6- Section 4.3 introduces different dropout schedules. My understanding is that "No Dropout" is ME-AL, "Full Dropout" is M--- or M--AL, and "Half Dropout" is MEDAL. What does "Probabilistic Dropout" correspond to? Are there any results with it? Looking at Figure 6 and the description of Equation 1, it seems Half-Dropout is used only. Or is the dropout probabilistic during training?

7 Figure 10c is hard to parse. I understand they are visuals from the environment, but I do not know how this contributes to the paper, because the visuals seem meaningless (because the depth information is

lost). Given this, I do not think 10 of these visuals are necessary and I would probably reduce it to just a few. But this is not a major concern for sure.

Reviewer #3 (Remarks to the Author):

Review of

Learning Few-Shot Imitation as Cultural Transmission

By Bhoopchand et al.

For Nature Comm.

This is a very interesting and well-written paper. There are several quite unique and intriguing aspects to this work, such as the computational implementation of the ZPD. There are some problems with the theoretical framework, and a few other minor issues, outlined below.

Though this paper is not the first to use the term ‘inheritance’ with respect to culture, this wording is sloppy and potentially misleading. The distinguishing feature of an inherited trait is that it is transmitted vertically (e.g., from parent to offspring) by way of a self-assembly code (such as DNA). It is therefore not obliterated at the end of a generation. In biology, inherited traits are contrasted with **acquired traits**, which are those that are not transmitted vertically by way of a self-assembly code; they are obtained over the course of the individual’s life, and lost at the end of the generation. For example, you don’t inherit your mother’s tattoo; it’s not a trait she was born with, she **acquired** it over her lifetime.

Culture doesn’t have this same distinction between inherited traits and acquired traits, because **no** traits are transmitted vertically by way of a self-assembly code. Thus, it isn’t correct to say that cultural traits are inherited, because they aren’t transmitted vertically by way of a self-assembly code. It is more correct to say that cultural traits are acquired, because they can be attained during their bearer’s lifetime, but even this can lead to confusion, since (unlike biological acquired traits) cultural traits are not wiped out at the end of a generation. For example, once someone came up with the idea of putting a handle on a cup, cups could forever after have handles.

In short, the term ‘inheritance’ is problematic when applied to culture. This is easily corrected by replacing each instance of the word ‘inherited’ with another word such as ‘acquired’ or better, ‘transmitted.’

The authors write in the abstract of the paper that inheritance – by which they mean cultural transmission -- “powers cumulative cultural evolution.” It is not accurate to say that cultural

transmission powers cultural evolution. Transmission ensures the perpetuation of fit variants once they've been found; it does not generate them. Thus, it does not power cumulative cultural evolution. This was shown way back in the 90s with the very first agent-based model of cumulative cultural evolution (Gabora, 1995): when the ratio of invention to imitation was 0:1 (i.e., lots of transmission but no creativity) there was no cumulative change at all. It was only by increasing this ratio -- injecting creativity into the system -- that cumulative cultural evolution was observed (although **too** much creativity decreased the rate of cumulative change). Indeed, many species exhibit cultural transmission - i.e., fit actions spread through a population -- but there is no cumulative change; thus, their culture does not evolve.

Similarly, the authors say in the abstract that imitation **expands** our skills, tools, knowledge, and so forth, but the word **expands** here is incorrect; imitation merely preserves that which already exists, it does not bring new ideas into existence.

The authors unquestioningly buy into the controversial theory that cultural evolution is Darwinian and that it works through natural selection. Darwin's theory provided a very successful resolution to the problem of how organisms **accumulate** adaptive change across lineages despite that traits acquired over their lifetimes are **eliminated** at the end of each generation. (What perplexed Darwin is; if you discard any acquired change at the end of each generation, how does change build up over time?) Darwin's genius was to take a population approach: fitter individuals leave more offspring, who share their traits, and thus over generations, the proportion of different traits increases across the population as a whole. However (as mentioned earlier), in cultural evolution, acquired traits are not lost at the end of each generation (to go back to the earlier example, once one cup had a handle, cups could forever after have handles; the acquired trait of possessing a handle is retained). Thus, cultural evolution does not actually face the problem that Darwin's theory was designed to solve.

Moreover, when we look a little deeper into the mechanisms underlying these two evolutionary processes, the application of a Darwinian framework to culture appears even more misguided. As originally pointed out by von Neumann (1966), and discussed subsequently by John Holland (1996), lack of transmission of acquired traits (the signature characteristic of a selectionist process) occurs when germ cells (such as sperm and eggs) are protected from environmental change due to their possessing a self-assembly code that is used in two distinct ways: (i) actively interpreted during development to generate a soma (body), and (ii) passively copied without interpretation during reproduction to generate germ cells (sperm and eggs). Cultural evolution does not involve a self-assembly code used in these two ways. See (Gabora & Steel, 2021) for more in depth discussion, as well as a non-Darwinian theory of how culture evolves. In short, if the authors are committed to interpreting their findings within a Darwinian framework, they should acknowledge these serious shortcomings and address how they can be overcome.

I think the authors should reword the phrasing of "This paves the way for cultural evolution as an algorithm for developing artificial general intelligence" and similar phrasing on p. 13. While it makes

sense that cultural evolution algorithms have a key role to play in the development of AGI, and this paper makes a valuable contribution to that, it doesn't follow that it "paves the way" to AGI. An AGI will have to be much more than a compendium of transmitted knowledge and imitated actions, and it will have to go further than the capacity to generalize its knowledge. It must possess agency and be self-organizing, self-sustaining, and self-mending, and even self-reproducing in a piecemeal manner through the sharing of knowledge and ideas, as opposed to reproducing all at once. (This, by the way, is part of the rationale for applying the formal framework of autocatalytic networks to cognition, because they provide a setting for modeling phase transitions in networks -- from discrete elements to various kinds of integrated wholes -- as new structure grows out of existing structure; in addition, they scale up, and can be studied using well-developed analytic methods.) That said, the fusion of AI and cultural evolutionary psychology is indeed a promising direction, and if the authors would care to speculate even further on such developments, I think readers would welcome this.

Curious to know why the rectangular obstacles have a strange 'pillow' shaped bump at one end.

A few more specific points, listed by page number.

p. 1 "A convincing demonstration of cultural transmission would..."

Why the conditional tense here? Are they implying that existing demonstrations of cultural transmission are not convincing? Cultural transmission models, in which you get cultural contagion without necessarily cumulative change, have a long history, going back to Hutchins and Hazelhurst (1990), and cumulative cultural evolution models just a few years later (as mentioned above), and much else has been done since. I actually think that what is done here goes well beyond what has been done before, but it would be helpful if the authors could articulate a little more precisely what is new and original about their contribution.

p. 2 – Re the challenges of solving correspondence problem –it might be appropriate to cite here Mitchell and Hofstadter's 'Copycat' which was the first of many computational models to address the challenge of "doing the same thing" given a different initial state.

p. 2 – My understanding is that it is still a matter of controversy to what extent the capacity to copy is hardwired or develops during childhood. While the *contents* of what gets imitated depends on culture, I doubt the *ability to imitate* is dependent on culture. See for instance the literature on neonatal imitation of facial gestures, and discussion in (Jones, 2017; Nagy et al, 2020). See also Roige & Carruthers' (2019) analysis of the 'cognitive gadget' thesis.

p. 2 – It would be helpful to have a clear sense up front about why expert dropout (D) is necessary. The other elements seem fairly intuitive or are clearly explained when first introduced, but it wasn't until several pages after expert dropout was first mentioned that I understood what it was.

p. 2 “We analyse the capabilities and limitations of our agent’s cultural transmission abilities on three axes inspired by the cognitive science of imitation, namely recall, generalisation, and fidelity.”

- It would be helpful to have the reference here.

p. 3 – “This train-test split provides evidence to rule out overfitting...” – Not completely sure what this is saying. Do you mean it provides data that enables overfitting to be ruled out?”

p. 5 – I see the value in distinguishing between following (just doing what someone successful is doing) and cultural transmission (remembering the socially learned behavior, and making use of it, after the teacher of it is gone). However, instead of saying “the more advanced social learning skill that we call cultural transmission” it would be better to do something that draws attention to the fact that you are using this term slightly differently than it has been used in the past, something along the lines of “in this paper, we...” and then italicize or bold the word so the reader can easily find it. (That goes for other terms too.) Or even better, a table defining all terms.

Re “We do not see this experimentation behaviour emerge in the absence of prior social learning abilities: social learning creates the right prior for experimentation to emerge.” True enough, but in real societies, while social learning is necessary, it's not sufficient. Innovators begin by learning the tools of the trade from others, i.e., by standing on the shoulders of giants, but few who stand on the shoulders of giants reach the magic beanstock.

P, 6 – Hard to distinguish btwn the 2 kinds of dotted green lines.

It isn't until the bottom of p. 6 that I completely understood the different test conditions. This information should come earlier in the paper.

p. 5 - The word 'together' implies two pieces of information are being considered, so instead of "this" the next word should be "these"

p. 13 Re “a selection process for knowledge.” Although the term ‘selection’ is used informally to mean ‘choosing,’ it also has a scientific meaning. In a scientific paper it seems prudent to use it only in its

scientific sense, where it refers to differential replication of randomly generated heritable variants in a population over generations, resulting in some traits becoming more prevalent than others. The scientific use of the word selection does not seem applicable here. First, knowledge is **not** randomly generated; it comes about through the strategic mental operations, and intuition, which reflects subconsciously detected patterns. Second, as mentioned above, cultural knowledge is not inherited. Third, culture does not exhibit the distinguishing characteristic of a selectionist process: lack of transmission of acquired traits across generations. Fourth, even if you come up with a rationale for what constitutes a population, and what constitutes a generation, since acquired traits are not wiped out each generation, and indeed new knowledge is being acquired each second, acquired traits accumulate so quickly they would render any change due to a natural selection type process (which requires generations) completely negligible. (Again, see Gabora & Steel, 2021 for details.)

p. 7 – “do cultural transmission in more complex tasks” – awkward phrasing; one doesn’t **do** cultural transmission. Perhaps something like “transmit cultural information in more complex tasks.”

p. 7 – I don’t think the Y axes of the centre and right graphs in 4b should both be labeled ‘Cultural Transmission.’ Perhaps put training CT on middle one and evaluation CT on right one?

p. 8 - Re “Within-episode recall from expert demonstrations has not previously been shown to arise from reinforcement learning in complex 3D environments,” – Can the authors clarify whether all of these aspects (e.g., Within-episode recall, expert demonstrations, RL, 3D envt) new, or just the conjunction of them new? Also, it’s good that this is new, but newness isn’t sufficient - can you add another sentence here saying why it matters?

p. 9 – caption: Instead of “Evidence of Causality” it would be better to say evidence that x causes y.

p. 13 – Re “the hypothesis that cultural evolution in a population of agents may yield progressively more generally capable artificial intelligence.” Yes, but simple demonstrations of this go back to the above-mentioned Meme and Variations program from the 90s, with subsequent more impressive demonstrations by that team and others. That said, I suspect that none of these efforts, including the work described in this paper, is but a tiny step toward artificial general intelligence, for reasons outlined above. So I suggest tempering this claim.

p. 14 – 1st para - remove ‘during’ where it says ‘during at’

p. 14 – Re “If that cultural transmission policy is robust, then RL must have favoured imitation with high fidelity, generalising across a range of contexts, and recalling transmitted behaviours.” – Generalization

across a range of contexts is not usually considered imitation, because it goes above and beyond the capacity to repeat what was observed. Moreover, if you generalize across contexts, you alter the imitated action, so fidelity is lower. So, this sentence needs some finessing.

p. 16 – comma after i.e.

Usually, the Discussion section comes after the Methods section.

A large fraction of the papers listed in the references are ArXiv preprints. If they have been published in journals, the journal info should be provided.

In summary, I believe the core of this paper provides a valuable contribution to the literature and is very worthy of publication. As written, there are a few flaws in the conceptual framework and conclusions, outlined above, but these problems are solvable.

Papers Cited

Gabora, L. (1995). Meme and variations: A computer model of cultural evolution. In (L. Nadel & D. Stein, Eds.) 1993 Lectures in Complex Systems (pp. 471-486). Boston: Addison-Wesley.
<http://arxiv.org/abs/1309.7524>

Gabora, L. & Steel, M. (2021). An evolutionary process without variation and selection. *Journal of the Royal Society Interface*, 18(180). 20210334.
<https://www.biorxiv.org/content/10.1101/2020.08.30.274407v1>

Holland J. H. (1992). *Adaptation in natural and artificial systems: an introductory analysis with applications to biology, control, and artificial intelligence*. Urbana, IL: MIT Press.

Hutchins, E., & Hazlehurst, B. (1990). *Learning in the cultural process*. Department of Cognitive Science, University of California, San Diego.

Jones, S. (2017). Can newborn infants imitate? *Wiley Interdisciplinary Reviews: Cognitive Science*, 8(1-2), e1410.

Mitchell, M. (1993) *Analogy-making as perception: A computer model*. ISBN 0-262-13289-3. MIT Press.

Nagy, E., Pilling, K., Blake, V., & Orvos, H. (2020). Positive evidence for neonatal imitation: A general response, adaptive engagement. *Developmental Science*, 23(2), e12894.

Roige, A., & Carruthers, P. (2019). Cognitive instincts versus cognitive gadgets: A fallacy. *Mind & Language*, 34(4), 540-550.

Von Neumann J. (1966). *Theory of self-replicating automata*. Urbana, IL: University of Illinois Press.

Reviewer #1 (Remarks to the Author):

This paper presents a machine learning study of cultural transmission, introducing an agent capable of sophisticated imitation, memory, and generalization. Through careful comparison with ablated agents, the authors establish the minimal conditions for effective cultural transmission in their space of agents. Finally, they peek inside the "brain" of their agent to identify neurons that code for significant social and goal variables.

Overall, I thought this was a very nice paper. It is clearly written, accessible to non-specialists, thorough, rigorous, and appealing to a broad range of readers (including cognitive scientists). In my view the paper could be accepted more or less as is. I also want to point out that I am not an expert in this area, so my evaluation should be considered with that in mind.

I have a few minor comments:

- I'm not really sure what is being learned from the "introspection" analysis (section 2.9). The authors find some neurons that code for particular things, but **it wasn't clear to me what that was telling us** (obviously the same criticisms apply to neuroscientists doing similar things!).

Deep learning models are not necessarily readily interpretable. On the other hand, interpretability is often desirable or even prerequisite for deploying AI systems in the real world (see [1] for a review). The purpose of this introspection analysis is to demonstrate that our model is interpretable at the neural level. In doing so, we also validate that the theoretical results in [2] yield interpretable sufficient statistics at the neural level even in tasks that are significantly more complex than were considered in that work.

We have rephrased the introduction to this section to better answer the question "why" this analysis is interesting.

[1] Linardatos, P., Papastefanopoulos, V., & Kotsiantis, S. (2020). Explainable AI: A review of machine learning interpretability methods. *Entropy*, 23(1), 18.

[2] Ortega, P. A., Wang, J. X., Rowland, M., Genewein, T., Kurth-Nelson, Z., Pascanu, R., ... & Legg, S. (2019). Meta-learning of sequential strategies. *arXiv preprint arXiv:1905.03030*.

- **Please make sure that the error bars are identified in the figure captions.** Also, sometimes standard deviation is reported, although standard error or 95% confidence intervals are more standard.

We have updated the captions of Figure 7 and 8b to identify the error bars.

- This is probably outside the scope of the paper, but as a cognitive scientist I would have been interested in the extent to which this approach could be used to model what we know about cultural transmission in humans. Furthermore, it would be interesting to investigate whether the ablated versions of the model correspond to the more limited forms of cultural transmission observed in non-human animals. **I think this might be worth at least a discussion point.**

We have inserted the following discussion point into the final paragraph, speculating on how our research might be useful as a model of the emergence of cultural transmission capabilities in humans, and as a means for analysing the ongoing process of cultural evolution in human society, particularly as AI becomes increasingly prevalent in our everyday lives.

"One could imagine comparing the various ablated models of MEDAL-ADR with the behaviour of children at different stages of ontogeny, or with the behaviour of non-human animals, were one to design appropriate tasks that control for the various sensorimotor differences between AIs, humans and animals. Alternatively, one might consider a line of experiments that investigated cultural accumulation across several ``generations" of humans and AIs in a laboratory environment, drawing comparisons between the different populations, or analysing the effects of mixing human and AI participants in a population."

Reviewer #2 (Remarks to the Author):

This paper proposes a method, called MEDAL-ADR, to enable artificial intelligence (AI) agents to achieve few-shot imitation. Humans are good at observing/following an expert performing a task for some short time and then imitating them to achieve the task even when the expert is not present anymore. The goal of this paper is to equip the AI agents with this ability as well.

The proposed solution relies on reinforcement learning (RL), but the actual contribution is not the RL algorithm. Instead, the contribution of the paper is the investigation of components that are useful for training e.g., the expert should sometimes disappear to discourage the agent from simply copying the expert. The paper is well-written in general, and extensive experiments have been presented.

Below are my comments to the authors.

1- In the paper, the authors acknowledge this work is only about imitation whereas cultural transmission can be in many other forms. Given this, **I think it is a little overselling to repeatedly mention and frame the work as achieving cultural transmission.** The term "few-shot imitation", which is what this paper is really about, is already descriptive and interesting to the community.

We thank the reviewer for their insightful comment. We agree with the reviewer that there is a potential for overclaiming, so we have clarified our use of the term "cultural transmission" in the introduction, indicating that our demonstration of cultural transmission is relatively narrow in this paper, restricted as it is to few-shot imitation. We expect, however, that our methods (such as expert dropout and the cultural transmission metric) would extend to a broader class of mechanisms for cultural transmission.

For the reviewer's context, our decision to use the term "cultural transmission" was mainly for clarity. There are already a number of works on "imitation learning" in the machine learning literature which take a quite different perspective on the problem. Imitation learning typically assumes a large, offline, first-person dataset of demonstrations, as opposed to a single, online, real-time, first-person demonstration. We have used "cultural transmission" as a shorthand that connects directly to cultural evolutionary psychology literature, and is not subject to confusion in the machine learning literature.

2- Another point that I think is overselling is that the paper states they identify the sufficient components of the training to achieve few-shot imitation, and it also mentions some of the components as necessary based on experiment results. These statements carry mathematical claims, but there is no evidence. First, **I disagree they are sufficient.** The accurate statement would be they are sufficient for this specific environment. Because there is currently no evidence that MEDAL-ADR can achieve few-shot imitation in other environments as well, despite the claims of the current environment being very general.

We have amended the introduction to clarify that we are only claiming sufficiency in the GoalCycle3D task space. We have also amended the discussion to point out more clearly that our minimal sufficient set is unlikely to be sufficient for all environments.

I also **disagree the components are necessary**. It might be possible that some other components (and their combinations) substitute the components of MEDAL-ADR that seems necessary in the experiments of this paper.

The reviewer is correct: we do not have any evidence for the necessity of these components. We have made changes to Section 2.4 and Section 2.6 to remove any hint that we are claiming necessity.

3- Why is the first term necessary / important in Equation 1? I think the second term is very well-motivated: the agent could learn simply copying the expert, so it is important to include a measure of success after the expert disappears. But it seems this second term already includes the cases where expert is present and absent. What does the first term contribute? Without the first term, a copycat agent (fully expert-dependent) would get a score of 0.5, which seems like a better scale.

Good question - this is a subtle point. It isn't quite true that the second term captures all the dynamics we are interested in. We would like the agent to be capable not only of remembering when the expert drops out halfway through the episode, but also of following the expert for the full episode when it is present for the full episode.

Therefore, having both terms actually gives us more signal than only having one of the terms. This is particularly informative early in training. For instance, the agent may start learning to follow in either the first half or the second half of the episode, in episodes where the expert is present throughout. Indeed the agent may be more likely to follow later in the episode, once it has had a chance to orient itself. The first term captures the possibility of following emerging in the second half of the episode, which would be absent if only the second term were used.

4- Why is this score decreasing in Figure 2? Is it because the environment is so easy that the agent is able to solve it on its own and the existence of the expert does not contribute much? In other words, is it easier / more efficient for the agent to explore on its own rather than waiting for the expert performing the task? Does this mean the expert is not playing the game perfectly and it is possible to outperform it with RL? If yes, why does the paper make this choice (of simple environment and imperfect expert)? Please clarify.

We have rewritten the last paragraph in Section 2.3 as follows, to provide more clarity:

"Lastly, in phase 4 (purple), the agent is able to solve the task independent of the expert bot. This is indicated by the training cultural transmission metric falling back towards 0 while the score continues to increase. The agent has learned a memory-based policy that can achieve high scores with or without the bot present. More precisely, MEDAL displays an "experimentation" behaviour in this phase, which involves using hypothesis-testing to infer the correct cycle without reference to the bot, followed by exploiting that correct cycle more efficiently than the bot does (see example videos). The bot is not quite optimal because for ease of programming it is hard-coded to pass through the centre of each correct goal sphere, whereas reward can be accrued by simply touching the sphere. Note by comparison with Figure 3a that this experimentation behaviour does not emerge in the absence of prior

social learning abilities. In other words, few-shot imitation creates the right prior for few-shot adaptation to emerge, which remarkably leads to improvement over the original demonstrator's policy."

To elaborate, at test time, the agent is performing no RL, it is rather executing a memory-dependent policy, which makes it capable of "in-context learning" (analogous to a large language model, for instance). This leads to two questions: (1) Can an agent do better than the expert with an optimal memory-based policy? (2) Can such a policy be learned with RL in the absence of cultural transmission?

The answer to (1), as the reviewer correctly interprets, is "yes, it can". This is what Phase 4 of Figure 2 shows. This is because the experts are slightly imperfect, as they are pre-programmed bots with a trajectory that visits the centre of every goal sphere. In fact, it's not necessary to visit the centre of the sphere to be rewarded, just to touch it, hence there is opportunity to outperform the expert. We made the choice for the expert to visit sphere centres because it was convenient to program, and in fact were happily surprised that the agent found this more efficient strategy, which would have been tricky to code.

The answer to (2) is "definitely not", as the green dotted line in Figure 3a shows. When training with pure RL, the agent is unable to discover a memory-based policy that can explore, discover the correct order of spheres and subsequently execute that correct order. This is because even this simple task is a very hard exploration problem for RL, due to the sparsity of reward and the inconsistency of the tasks between different episodes. Indeed, black-box meta-RL is known to struggle to learn at all in such settings (see for instance [3]).

This makes the behaviour learned in Phase 4 of our training all the more remarkable. We find that an agent with prior few-shot imitation ability is able to discover the optimal solo memory-based policy via RL. In other words, if one were to "luckily" initialise the agent to be capable of cultural transmission, then under those circumstances, RL would be enough to discover the "experiment, discover and exploit" policy we see in Phase 4. Of course, it's vanishingly unlikely that one initialises an agent from scratch that is capable of cultural transmission. We summarise this observation in the sentence "few-shot imitation creates the right prior for few-shot adaptation to emerge".

[3] Kirsch, L., Flennerhag, S., van Hasselt, H., Friesen, A., Oh, J., & Chen, Y. (2022, June). Introducing symmetries to black box meta reinforcement learning. In Proceedings of the AAAI Conference on Artificial Intelligence (Vol. 36, No. 7, pp. 7202-7210).

5- Discussion says "an agent could be trained with a variety of experts, including noisy experts and experts with objectives that are different from the agent's." **This would only make sense if the reward signal is part of the observation that the policy depends on, right?**

The reviewer's intuition here is correct, in the following sense. If the agent were to be trained with a heterogeneous mix of experts, for instance with a population of varying noise or varying objectives, then the agent must have a means of extracting the useful information from each expert on the fly. There are two potential solutions here:

(1) The experts have some consistent labels which the agent can learn to identify. This could be, for instance, an indication of their objective or noise level, or some derived measure of the difference between their behaviour and the desired behaviour of the agent. Such labels are not uncommon in nature, think of reputation and honest signalling for example [4].

(2) The agent must be able to derive, for itself, the relevance of the expert's behaviour for its own purposes, without any consistent labelling scheme for experts being provided. In this situation, as the reviewer rightly remarks, it is important for the agent to observe its own reward, in order to assess to what extent the expert's behaviour matches or mismatches with its own goal. Note that even when the expert's behaviour is exactly the opposite of the agent's desired outcome, it can still confer useful information: to take an ethological example, you can learn where the predator is by seeing a fellow prey animal getting caught.

[4] Számadó, S., Balliet, D., Giardini, F., Power, E. A., & Takács, K. (2021). The language of cooperation: reputation and honest signalling. *Philosophical Transactions of the Royal Society B*, 376(1838), 20200286.

6- Section 4.3 introduces different dropout schedules. My understanding is that "No Dropout" is ME-AL, "Full Dropout" is M--- or M--AL, and "Half Dropout" is MEDAL. **What does "Probabilistic Dropout" correspond to? Are there any results with it? Looking at Figure 6 and the description of Equation 1, it seems Half-Dropout is used only. Or is the dropout probabilistic during training?**

The reviewer is exactly correct to distinguish between the dropout used in training and the dropout used in evaluation.

For evaluation, we always generate scores in scenarios involving "full", "half" and "no" dropout, in order to calculate the CT metric. This is done regardless of the dropout scheme with which the agent was trained.

During training, we use a variety of different dropout schemes, depending on the ablation. M— is trained with full dropout (expert is never present), ME-AL is trained with no dropout (expert is always present) and all other agents are trained with probabilistic dropout.

We thank the reviewer for bringing this to our attention and have added the previous paragraph to the main text.

Figure 10c is hard to parse. I understand they are visuals from the environment, but I do not know how this contributes to the paper, because the visuals seem meaningless (because the depth information is lost). Given this, **I do not think 10 of these visuals are necessary and I would probably reduce it to just a few**. But this is not a major concern for sure.

Thank you for the suggestion. We have reduced the number of visuals to four per subfigure which we believe is enough to still highlight the diversity of our evaluation setup.

Reviewer #3 (Remarks to the Author):

We are particularly grateful to this reviewer for their deep expertise in the cultural evolution literature, and for pointing out several areas in which our language or exposition should be more precise. In summary, we have redrafted the text to take into account these suggestions, thereby more accurately situating our work in the context of the wider field. We now address each comment in detail.

This is a very interesting and well-written paper. There are several quite unique and intriguing aspects to this work, such as the computational implementation of the ZPD. There are some problems with the theoretical framework, and a few other minor issues, outlined below.

Though this paper is not the first to use the term 'inheritance' with respect to culture, this wording is sloppy and potentially misleading. The distinguishing feature of an inherited trait is that it is transmitted vertically (e.g., from parent to offspring) by way of a self-assembly code (such as DNA). It is therefore not obliterated at the end of a generation. In biology, inherited traits are contrasted with *acquired traits*, which are those that are not transmitted vertically by way of a self-assembly code; they are obtained over the course of the individual's life, and lost at the end of the generation. For example, you don't inherit your mother's tattoo; it's not a trait she was born with, she *acquired* it over her lifetime.

Culture doesn't have this same distinction between inherited traits and acquired traits, because *no* traits are transmitted vertically by way of a self-assembly code. Thus, it isn't correct to say that cultural traits are inherited, because they aren't transmitted vertically by way of a self-assembly code. It is more correct to say that cultural traits are acquired, because they can be attained during their bearer's lifetime, but even this can lead to confusion, since (unlike biological acquired traits) cultural traits are not wiped out at the end of a generation. For example, once someone came up with the idea of putting a handle on a cup, cups could forever after have handles.

In short, the term 'inheritance' is problematic when applied to culture. This is easily corrected by replacing each instance of the word 'inherited' with another word such as 'acquired' or better, 'transmitted.'

We thank the reviewer for pointing this out. We have checked the amended manuscript and there are no longer any instances of the word "inherited". Where we have used the word "inheritance", we have qualified that this is one way of thinking about the role of cultural transmission, thereby softening our claim.

The authors write in the abstract of the paper that inheritance – by which they mean cultural transmission -- “powers cumulative cultural evolution.” **It is not accurate to say that cultural transmission powers cultural evolution.** Transmission ensures the perpetuation of fit variants once they've been found; it does not generate them. Thus, it does not power cumulative cultural evolution. This was shown way back in the 90s with the very first agent-based model of cumulative cultural evolution (Gabora, 1995): when the ratio of invention to imitation was 0:1 (i.e., lots of transmission but no creativity) there was no cumulative change at all. It was only by increasing this ratio -- injecting creativity into the system -- that cumulative cultural evolution was observed (although *too* much creativity decreased the rate of cumulative change). Indeed, many species exhibit cultural

transmission -- i.e., fit actions spread through a population -- but there is no cumulative change; thus, their culture does not evolve.

We have rephrased the abstract to read "It can be thought of as the inheritance process that perpetuates fit variants in cultural evolution".

Similarly, the **authors say in the abstract that imitation *expands* our skills, tools, knowledge, and so forth, but the word *expands* here is incorrect**; imitation merely preserves that which already exists, it does not bring new ideas into existence.

We have rephrased the abstract to clarify that (cumulative) cultural evolution is responsible for the expansion in skills, tools and knowledge, not merely imitation.

The authors unquestioningly buy into the controversial theory that cultural evolution is Darwinian and that it works through natural selection. Darwin's theory provided a very successful resolution to the problem of how organisms *accumulate* adaptive change across lineages despite that traits acquired over their lifetimes are *eliminated* at the end of each generation. (What perplexed Darwin is; if you discard any acquired change at the end of each generation, how does change build up over time?) Darwin's genius was to take a population approach: fitter individuals leave more offspring, who share their traits, and thus over generations, the proportion of different traits increases across the population as a whole. However (as mentioned earlier), in cultural evolution, acquired traits are not lost at the end of each generation (to go back to the earlier example, once one cup had a handle, cups could forever after have handles; the acquired trait of possessing a handle is retained). Thus, cultural evolution does not actually face the problem that Darwin's theory was designed to solve.

Moreover, when we look a little deeper into the mechanisms underlying these two evolutionary processes, the application of a Darwinian framework to culture appears even more misguided. As originally pointed out by von Neumann (1966), and discussed subsequently by John Holland (1996), lack of transmission of acquired traits (the signature characteristic of a selectionist process) occurs when germ cells (such as sperm and eggs) are protected from environmental change due to their possessing a self-assembly code that is used in two distinct ways: (i) actively interpreted during development to generate a soma (body), and (ii) passively copied without interpretation during reproduction to generate germ cells (sperm and eggs). Cultural evolution does not involve a self-assembly code used in these two ways. See (Gabora & Steel, 2021) for more in depth discussion, as well as a non-Darwinian theory of how culture evolves. In short, **if the authors are committed to interpreting their findings within a Darwinian framework, they should acknowledge these serious shortcomings and address how they can be overcome.**

We thank the reviewer for bringing this controversy to our attention: we were not aware of these interesting intricacies!

We were hoping to appeal to the concept of evolution here in a more general sense than a strict analogy with biological (Darwinian) evolution might allow, i.e. along the lines of universal Darwinism [5]. This slightly more "free and easy" perspective is somewhat

common in machine learning, for instance as used in replicator dynamics [6] and population-based-training [7].

We have clarified in the introduction that we are framing our work in the context of a "universal Darwinian" interpretation of cultural evolution. In the discussion we have added a note to point out that alternative (non-Darwinian) interpretations of cultural evolution are available, including ones which don't involve variation and selection, such as [8]. Indeed our results remain relevant in this context, because fast acquisition / transmission of cultural traits is required for any account of cultural evolution.

[5] Dennett, D. C. (1995). Darwin's dangerous idea. *The Sciences*, 35(3), 34-40.

[6] Börgers, T., & Sarin, R. (1997). Learning through reinforcement and replicator dynamics. *Journal of economic theory*, 77(1), 1-14.

[7] Jaderberg, M., Dalibard, V., Osindero, S., Czarnecki, W. M., Donahue, J., Razavi, A., ... & Kavukcuoglu, K. (2017). Population based training of neural networks. arXiv preprint arXiv:1711.09846.

[8] Gabora, L., & Steel, M. (2021). An evolutionary process without variation and selection. *Journal of the Royal Society Interface*, 18(180), 20210334.

I think the authors should **reword the phrasing of “This paves the way for cultural evolution as an algorithm for developing artificial general intelligence” and similar phrasing on p. 13**. While it makes sense that cultural evolution algorithms have a key role to play in the development of AGI, and this paper makes a valuable contribution to that, it doesn't follow that it “paves the way” to AGI. An AGI will have to be much more than a compendium of transmitted knowledge and imitated actions, and it will have to go further than the capacity to generalize its knowledge. It must possess agency and be self-organizing, self-sustaining, and self-mending, and even self-reproducing in a piecemeal manner through the sharing of knowledge and ideas, as opposed to reproducing all at once. (This, by the way, is part of the rationale for applying the formal framework of autocatalytic networks to cognition, because they provide a setting for modeling phase transitions in networks -- from discrete elements to various kinds of integrated wholes -- as new structure grows out of existing structure; in addition, they scale up, and can be studied using well-developed analytic methods.)

We have amended the final sentence of the abstract to read: "This paves the way for cultural evolution to play an algorithmic role in the development of AGI". We have also made a number of rewrites in the Discussion section which we hope softens our claim appropriately.

That said, the fusion of AI and cultural evolutionary psychology is indeed a promising direction, and **if the authors would care to speculate even further on such developments, I think readers would welcome this**.

We have added a few sentences of speculation to the final paragraph of the Discussion section.

Curious to know why the rectangular obstacles have a strange ‘pillow’ shaped bump at one end.

Good question! In an early iteration of the environment styling, we were inspired by naturalistic scenarios like trees in a forest. The mesh shapes for the obstacles were chosen with this in mind: think "fallen tree trunks". Shortly after, we decided that a more colourful and abstract visual styling would be more appropriate, but we kept the mesh shape the same.

A few more specific points, listed by page number.

p. 1 "A convincing demonstration of cultural transmission would..."

Why the conditional tense here? Are they implying that existing demonstrations of cultural transmission are not convincing? Cultural transmission models, in which you get cultural contagion without necessarily cumulative change, have a long history, going back to Hutchins and Hazelhurst (1990), and cumulative cultural evolution models just a few years later (as mentioned above), and much else has been done since. I actually think that what is done here goes well beyond what has been done before, but **it would be helpful if the authors could articulate a little more precisely what is new and original about their contribution.**

This is a great point: our use of the conditional tense was inappropriate here. We have amended this section of the introduction as follows, to more clearly point out the novelty of our contribution.

"By exhibiting cultural transmission among embodied artificial agents in a complex space of 3D interactive tasks, we extend a previous literature on computational models of cultural evolution (going as far back as [9]) in the direction of using cultural evolution as an AI-generating algorithm [10, 11]."

[9] Hutchins, E., & Hazlehurst, B. (1990). Learning in the cultural process. Department of Cognitive Science, University of California, San Diego.

[10] Clune, J. (2019). AI-GAs: AI-generating algorithms, an alternate paradigm for producing general artificial intelligence. arXiv preprint arXiv:1905.10985.

[11] Leibo, J. Z., Hughes, E., Lanctot, M., & Graepel, T. (2019). Autocurricula and the emergence of innovation from social interaction: A manifesto for multi-agent intelligence research. arXiv preprint arXiv:1903.00742.

p. 2 – Re the challenges of solving correspondence problem –it might be appropriate to cite here Mitchell and Hofstadter’s ‘Copycat’ which was the first of many computational models to address the challenge of “doing the same thing” given a different initial state.

Thank you, we have inserted this citation.

p. 2 – My understanding is that it is still a matter of controversy to what extent the capacity to copy is hardwired or develops during childhood. While the *contents* of what gets imitated depends on culture, I doubt the *ability to imitate* is dependent on culture. See for instance the literature on neonatal imitation of facial gestures, and discussion in (Jones, 2017; Nagy et al, 2020). See also Roige & Carruthers’ (2019) analysis of the ‘cognitive gadget’ thesis.

Thank you for pointing out these references: we agree that our original statement was too strong. We have edited this sentence to read:

"We want to know how the skill of cultural transmission can develop when an artificial agent is learning from scratch, analogous in the cognitive science literature to the combination of phylogeny and ontogeny."

p. 2 – **It would be helpful to have a clear sense up front about why expert dropout (D) is necessary.** The other elements seem fairly intuitive or are clearly explained when first introduced, but it wasn't until several pages after expert dropout was first mentioned that I understood what it was.

We have inserted the following clarifying sentence to the penultimate paragraph of the introduction:

"This probabilistic dropout provides the right experience for agents to learn to observe what a useful demonstrator is doing and then remember and reproduce it when the demonstrator is absent."

p. 2 "We analyse the capabilities and limitations of our agent's cultural transmission abilities on three axes inspired by the cognitive science of imitation, namely recall, generalisation, and fidelity."

- **It would be helpful to have the reference here.**

We have inserted appropriate references.

p. 3 – "This train-test split provides evidence to rule out overfitting..." – Not completely sure what this is saying. **Do you mean it provides data that enables overfitting to be ruled out?"**

Yes, precisely. We have amended the sentence to make this clear.

p. 5 – I see the value in distinguishing between following (just doing what someone successful is doing) and cultural transmission (remembering the socially learned behavior, and making use of it, after the teacher of it is gone). However, instead of saying "the more advanced social learning skill that we call cultural transmission" **it would be better to do something that draws attention to the fact that you are using this term slightly differently than it has been used in the past, something along the lines of "in this paper, we..." and then italicize or bold the word so the reader can easily find it.** (That goes for other terms too.) Or even better, a table defining all terms.

We have completely rewritten the start of Section 2.2, italicising "cultural transmission" and being more precise about exactly how we are choosing to define this term.

We have also added Appendix A which provides a glossary of terms, as follows:

A *policy* for an agent is a distribution over actions for each state in the environment. In our setting, the actions that can be taken are physical movement, so a policy represents a physical movement behaviour.

Cultural transmission from expert to agent is defined to be the extent to which the agent can improve its score during and after the expert has been present in the world. See Section 2.2 for a mathematical formalisation of this.

Imitation is the ability of an agent to copy salient features of the policy of a third-person demonstrator in real time. This is known as few-shot, embodied, third-person imitation in the machine learning literature. It is a particular (high-fidelity) kind of cultural transmission.

Adaptation is the ability of an agent to improve its policy online, on-the-fly and from a single stream of experience based on the discovery of new information. In our setting, adaptation is typically made possible by an agent's internal memory.

Social learning is the process by which cultural transmission occurs; i.e., an individual's acquisition of previously unknown information or behaviour by observing another individual in real time. It is a kind of adaptation facilitated by the presence of a co-player in the environment.

The *fidelity* of cultural transmission is the extent to which the (score-salient) behaviour of the expert can be reproduced by the agent without mistakes. In our setting, this corresponds to the agent reliably reproducing the same correct patterns of sphere visits as the expert. \

Generalisation is an agent's ability to successfully demonstrate cultural transmission across a wide variety of different environmental conditions, including in environment variants not seen during training.

Recall is the ability of an agent to reproduce the (score-salient) behaviour of an expert after that expert has departed from the world.

The *robustness* of a policy is a shorthand for referring to the policy's generalisation, fidelity and recall abilities.

A *probe task* is a hand-picked evaluation task which is (probabilistically) held-out from training, and used to assess the robustness of cultural transmission. Probe tasks are held fixed so we can get a fair comparison across agents ablated in different ways.

Expert dropout is a training procedure in which each episode of experience for the agent consists of some period(s) in which the expert is present and some period(s) in which the expert is absent ("dropped out").

Automatic domain randomization is a training procedure in which each episode of experience for an agent consists of a task sampled from a distribution, the parameters of which gradually change over the course of training, based on the value of a metric (in our case, cultural transmission).

Re “We do not see this experimentation behaviour emerge in the absence of prior social learning abilities: social learning creates the right prior for experimentation to emerge.” **True enough, but in real societies, while social learning is necessary, it’s not sufficient.** Innovators begin by learning the tools of the trade from others, i.e., by standing on the shoulders of giants, but few who stand on the shoulders of giants reach the magic beanstock.

We agree with the reviewer and have added the following sentence to make this clear: "Note that, social learning by itself is not enough to generate experimentation automatically, further innovation by reinforcement learning, on top of the culturally transmitted prior, is necessary for the agent to exceed the capabilities of its expert partner. Our agent stands on the shoulders of giants, and then riffs to climb yet higher."

P, 6 – **Hard to distinguish btwn the 2 kinds of dotted green lines.**

Throughout the manuscript, we have aimed to keep a consistent colour for each variation of our agents. As this is hard to distinguish in this plot, we have provided the raw data in Source Data.

It isn’t until the bottom of p. 6 that I completely understood the different test conditions. **This information should come earlier in the paper.**

We have added the following sentence to the penultimate paragraph of the introduction to foreshadow these ablations: "These components are ablated in turn in Sections 2.4 to 2.6: only when all of them are acting in concert does robust cultural transmission arise in complex worlds."

p. 5 - **The word 'together' implies two pieces of information are being considered, so instead of "this" the next word should be "these"**

We have rewritten this paragraph.

p. 13 Re “a selection process for knowledge.” Although the term ‘selection’ is used informally to mean ‘choosing,’ it also has a scientific meaning. In a scientific paper it seems prudent to use it only in its scientific sense, where it refers to differential replication of randomly generated heritable variants in a population over generations, resulting in some traits becoming more prevalent than others. **The scientific use of the word selection does not seem applicable here.** First, knowledge is *not* randomly generated; it comes about through the strategic mental operations, and intuition, which reflects subconsciously detected patterns. Second, as mentioned above, cultural knowledge is not inherited. Third, culture does not exhibit the distinguishing characteristic of a selectionist process: lack of transmission of acquired traits across generations. Fourth, even if you come up with a rationale for what constitutes a population, and what constitutes a generation, since acquired traits are not wiped out each generation, and indeed new knowledge is being acquired each second, acquired traits accumulate so quickly they would render any change due to a natural selection type process (which requires generations) completely negligible. (Again, see Gabora & Steel, 2021 for details.)

We have edited this paragraph (and the introduction) to make it clear that we are framing cultural evolution in a "universal Darwinian" sense (as in, for example, [12]). We have also added a sentence referencing [8], to point out that other framings for cultural evolution are available in the literature.

[8] Gabora, L., & Steel, M. (2021). An evolutionary process without variation and selection. *Journal of the Royal Society Interface*, 18(180), 20210334.

[12] Blackmore, S. (2000). *The meme machine* (Vol. 25). Oxford Paperbacks.

p. 7 – “do cultural transmission in more complex tasks” – awkward phrasing; one doesn’t *do* cultural transmission. **Perhaps something like “transmit cultural information in more complex tasks.”**

We have made this change.

p. 7 – I don’t think the Y axes of the centre and right graphs in 4b should both be labeled ‘Cultural Transmission.’ **Perhaps put training CT on middle one and evaluation CT on right one?**

Thank you for the suggestion. We have changed the Y-axis in Figure 3a, 3b, and 4b to all say “Training CT” in the middle graph and “Evaluation CT” on the right axis.

p. 8 - Re “Within-episode recall from expert demonstrations has not previously been shown to arise from reinforcement learning in complex 3D environments,” – **Can the authors clarify whether all of these aspects (e.g., Within-episode recall, expert demonstrations, RL, 3D envt) new, or just the conjunction of them new? Also, it’s good that this is new, but newness isn’t sufficient - can you add another sentence here saying why it matters?**

We have amended this sentence, replacing it with the following, more detailed, explanation of the novelty in our work and its significance for the field.

To our knowledge, within-episode recall of a third-person demonstration has not previously been shown to arise from reinforcement learning. This is an important discovery, since the recent history of AI research has demonstrated the increased flexibility and generality of learned behaviours over pre-programmed ones. What’s more, third-person recall within an episode amortises imitation onto a timescale of seconds and does not require perspective matching between co-players. As such, we achieve the fast adaptation benefits of previous first-person few-shot imitation works (e.g. [13-15]) but as a general-purpose emergent property from third-person RL rather than via a special-purpose first-person imitation algorithm.

[13] Edwards, A., Sahni, H., Schroecker, Y., & Isbell, C. (2019, May). Imitating latent policies from observation. In *International conference on machine learning* (pp. 1755-1763). PMLR.

[14] James, S., Bloesch, M., & Davison, A. J. (2018, October). Task-embedded control networks for few-shot imitation learning. In *Conference on robot learning* (pp. 783-795). PMLR.

[15] Duan, Y., Andrychowicz, M., Stadie, B., Jonathan Ho, O., Schneider, J., Sutskever, I., ... & Zaremba, W. (2017). One-shot imitation learning. *Advances in neural information processing systems*, 30.

p. 9 – caption: Instead of “Evidence of Causality” it would be better to say evidence that x causes y.

This is hard to do while also remaining concise. For instance, "evidence that the expert's demonstration causes the agent to follow the same path" is too wordy to be a good paragraph heading. Since this is explained in more detail in the paragraph itself, we propose to leave this unchanged.

p. 13 – Re “the hypothesis that cultural evolution in a population of agents may yield progressively more generally capable artificial intelligence.” Yes, but simple demonstrations of this go back to the above-mentioned Meme and Variations program from the 90s, with subsequent more impressive demonstrations by that team and others. That said, I suspect that none of these efforts, including the work described in this paper, is but a tiny step toward artificial general intelligence, for reasons outlined above. So **I suggest tempering this claim.**

We agree that this is just another step along a (potentially long) road. Therefore, we have made this hypothesis more specific, writing:

"It would be fascinating to validate or falsify the hypothesis that cultural evolution in a population of agents may lead to the accumulation of behaviours that solve an ever-wider array of human-relevant real-world problems."

We hope that this will serve as a "call to arms" for researchers in the machine learning space to pay attention to the rich literature in cultural evolution which the reviewer has mentioned.

p. 14 – 1st para - remove ‘during’ where it says ‘during at’

We have made this correction.

p. 14 – Re “If that cultural transmission policy is robust, then RL must have favoured imitation with high fidelity, generalising across a range of contexts, and recalling transmitted behaviours.” – Generalization across a range of contexts is not usually considered imitation, because it goes above and beyond the capacity to repeat what was observed. Moreover, if you generalize across contexts, you alter the imitated action, so fidelity is lower. So, **this sentence needs some finessing.**

What we mean here is that the ability to imitate shouldn't depend on the specific context in which the imitation occurs, for instance the precise layout of obstacles and relative positions of the agent and expert. To make this clearer we have changed this sentence to read as follows.

"If that cultural transmission policy is robust, then RL must have favoured imitation with high fidelity, which generalises across a range of physical contexts, and where transmitted behaviours are recalled after the demonstrator has departed."

p. 16 – comma after i.e.

We have made this correction.

Usually, the Discussion section comes after the Methods section.

We have swapped the order of the Discussion and Methods sections, as suggested.

A large fraction of the papers listed in the references are ArXiv preprints. **If they have been published in journals, the journal info should be provided.**

Thank you for pointing this out. We have made a thorough pass through our references and added journal information for all papers where this is possible. Since the references are automatically generated by LaTeX, our changes are not shown in blue, but they should readily be apparent by comparison to the previous version.

There remain a few preprints in the references: it is fairly common in the Machine Learning field for some works to appear only as preprints.

In summary, I believe the core of this paper provides a valuable contribution to the literature and is very worthy of publication. As written, there are a few flaws in the conceptual framework and conclusions, outlined above, but these problems are solvable.

Papers Cited

Gabora, L. (1995). Meme and variations: A computer model of cultural evolution. In (L. Nadel & D. Stein, Eds.) 1993 Lectures in Complex Systems (pp. 471-486). Boston: Addison-Wesley. <http://arxiv.org/abs/1309.7524>

Gabora, L. & Steel, M. (2021). An evolutionary process without variation and selection. *Journal of the Royal Society Interface*, 18(180). 20210334. <https://www.biorxiv.org/content/10.1101/2020.08.30.274407v1>

Holland J. H. (1992). *Adaptation in natural and artificial systems: an introductory analysis with applications to biology, control, and artificial intelligence*. Urbana, IL: MIT Press.

Hutchins, E., & Hazlehurst, B. (1990). Learning in the cultural process. Department of Cognitive Science, University of California, San Diego.

Jones, S. (2017). Can newborn infants imitate? *Wiley Interdisciplinary Reviews: Cognitive Science*, 8(1-2), e1410.

Mitchell, M. (1993) *Analogy-making as perception: A computer model*. ISBN 0-262-13289-3. MIT Press.

Nagy, E., Pilling, K., Blake, V., & Orvos, H. (2020). Positive evidence for neonatal imitation: A general response, adaptive engagement. *Developmental Science*, 23(2), e12894.

Roige, A., & Carruthers, P. (2019). Cognitive instincts versus cognitive gadgets: A fallacy. *Mind & Language*, 34(4), 540-550.

Von Neumann J. (1966). *Theory of self-replicating automata*. Urbana, IL: University of Illinois Press.

REVIEWER COMMENTS

Reviewer #1 (Remarks to the Author):

I am satisfied with the revisions made in response to my comments.

Reviewer #2 (Remarks to the Author):

I thank the authors for their extensive response to my previous comments. They addressed most of my concerns, although I still believe the term "few-shot imitation" is more suitable than the more general "cultural transmission". Given the other reviewers were not concerned about this, I will assume it was just my personal taste.

I suggest using "artificial general intelligence" in its expanded form first, before using the acronym AGI.

Reviewer #3 (Remarks to the Author):

Review for Nature Communications

Manuscript#: NCOMMS-22-52064A

Corresponding Author: Edward Hughes

Title: Learning Few-Shot Imitation as Cultural Transmission

The revisions have improved the manuscript, though some problems remain.

The authors misread the earlier comment about the history of computer models of cultural evolution. The first computer model of cultural evolution is:

Gabora, L. (1995). Meme and variations: A computer model of cultural evolution. In (L. Nadel & D. Stein, Eds.) 1993 Lectures in Complex Systems (pp. 471-486). Boston: Addison-Wesley.
<http://arxiv.org/abs/1309.7524>

Hutchins and Hazelhurst's model was not of cultural evolution. It was a model of cultural transmission (i.e., multiple variants that converge on one or the other), not cultural evolution, since evolution requires cumulative change, and their model had no cumulative change.

The authors have not actually addressed the critique in the previous review by framing their analysis in terms of universal Darwinism, since clearly this problem applies equally well to universal Darwinism. Their response is basically 'other lemmings have jumped off the cliff, so we are too.' My sense is that the authors may have not digested the extent of the problem (they write about an "alternative" view "without variation and selection" but if they read beyond the title of the paper they are referring to here they will see that it is only in a very special case that no variation is involved). It is unscientific to pretend to live in a world in which such a fundamental problem with their chosen framework has never been brought to light. If they are insisting on framing their work in terms of universal Darwinism, given that serious problems with this framework have been identified in the published literature on this topic (as laid out in quite some detail in my original review), this choice should either be justified, or changed. I note that there is no reason this work needs to be in terms of universal Darwinism, as it can equally well be interpreted from within a non-Darwinian framework.

It isn't clear why (even the 'softened') misuse of the term inheritance is necessary when more accurate alternatives (such as transmission) exist. In the abstract and on p. 1 could be made more correct by simply removing the word 'inheritance' from the phrase 'inheritance process.'

Reviewer #1 (Remarks to the Author):

I am satisfied with the revisions made in response to my comments.

We thank the reviewer again for their comments.

Reviewer #2 (Remarks to the Author):

I thank the authors for their extensive response to my previous comments. They addressed most of my concerns, although I still believe the term "few-shoot imitation" is more suitable than the more general "cultural transmission". Given the other reviewers were not concerned about this, I will assume it was just my personal taste.

I suggest using "artificial general intelligence" in its expanded form first, before using the acronym AGI.

We thank the reviewer for pointing this out: we have expanded the acronym to "artificial general intelligence" where it appears.

We thank the reviewer again for their comments.

Reviewer #3 (Remarks to the Author):

The revisions have improved the manuscript, though some problems remain.

The authors misread the earlier comment about the history of computer models of cultural evolution. The first computer model of cultural evolution is:

Gabora, L. (1995). Meme and variations: A computer model of cultural evolution. In (L. Nadel & D. Stein, Eds.) 1993 Lectures in Complex Systems (pp. 471-486). Boston: Addison-Wesley. <http://arxiv.org/abs/1309.7524>

Hutchins and Hazlehurst's model was not of cultural evolution. It was a model of cultural transmission (i.e., multiple variants that converge on one or the other), not cultural evolution, since evolution requires cumulative change, and their model had no cumulative change.

We apologise for this misunderstanding, and are very grateful to the reviewer for the clarification. We have amended the relevant section of the introduction to read:

"By exhibiting cultural transmission among embodied artificial agents in a complex space of 3D interactive tasks, we extend a previous literature on computational models of cultural transmission [1] and cultural evolution [2] in the direction of using cultural evolution as an AI-generating algorithm [3,4]."

[1] Hutchins, E., & Hazlehurst, B. (1990). Learning in the cultural process. Department of Cognitive Science, University of California, San Diego.

[2] Gabora, L. (1995). Meme and variations: A computer model of cultural evolution. In (L. Nadel & D. Stein, Eds.) 1993 Lectures in Complex Systems (pp. 471-486). Boston: Addison-Wesley. <http://arxiv.org/abs/1309.7524>

[3] Clune, J. (2019). AI-GAs: AI-generating algorithms, an alternate paradigm for producing general artificial intelligence. arXiv preprint arXiv:1905.10985.

[4] Leibo, J. Z., Hughes, E., Lanctot, M., & Graepel, T. (2019). Autocurricula and the emergence of innovation from social interaction: A manifesto for multi-agent intelligence research. arXiv preprint arXiv:1903.00742.

The authors have not actually addressed the critique in the previous review by framing their analysis in terms of universal Darwinism, since clearly this problem applies equally well to universal Darwinism. Their response is basically ‘other lemmings have jumped off the cliff, so we are too.’ My sense is that the authors may have not have digested the extent of the problem (they write about an “alternative” view “without variation and selection” but if they read beyond the title of the paper they are referring to here they will see that it is only in a very special case that no variation is involved). It is unscientific to pretend to live in a world in which such a fundamental problem with their chosen framework has never been brought to light. If they are insisting on framing their work in terms of universal Darwinism, given that serious problems with this framework have been identified in the published literature on this topic (as laid out in quite some detail in my original review), this choice should either be justified, or changed. I note that there is no reason this work needs to be in terms of universal Darwinism, as it can equally well be interpreted from within a non-Darwinian framework.

It isn't clear why (even the ‘softened’) misuse of the term inheritance is necessary when more accurate alternatives (such as transmission) exist. In the abstract and on p. 1 could be made more correct by simply removing the word ‘inheritance’ from the phrase ‘inheritance process.’

We thank the reviewer for their cogent argument. We agree that there is no need to view our work through a Darwinian lens. On deeper reading of the literature suggested by the reviewer, we now understand better how this is misleading. We apologise that we did not fully appreciate this subtlety in the first round of revisions.

We have therefore decided to remove the framing in terms of universal Darwinism. Explicitly, we have made the following changes to the manuscript:

- In the abstract and the introduction we have removed the word "inheritance".
- In the introduction, we have removed the sentence "In a universal Darwinian framing, cultural evolution is portrayed as an inevitable consequence of three processes: variation, selection and inheritance."
- We have removed any mention of the term "inheritance" in the Appendix.
- In the discussion, we have removed mention of "inheritance", modifying the relevant section to read:

"Finally, cultural transmission is a necessary condition for generating cultural evolution but may or may not be sufficient (see [5] for a discussion). Earlier, we argued that appropriate randomisation over experts may generate a selection process for knowledge. One might

also ask for variation in behaviour space to power the evolutionary system. Fortunately, there are a variety of off-the-shelf techniques for generating diverse policies [6,7]."

[5] Gabora, L. & Steel, M. (2021). An evolutionary process without variation and selection. *Journal of the Royal Society Interface*, 18(180). 20210334.

<https://www.biorxiv.org/content/10.1101/2020.08.30.274407v1>

[6] Rakicevic, Nemanja, Antoine Cully, and Petar Kormushev. "Policy manifold search: Exploring the manifold hypothesis for diversity-based neuroevolution." *Proceedings of the Genetic and Evolutionary Computation Conference*. 2021.

[7] Eysenbach, B., Gupta, A., Ibarz, J., & Levine, S. (2018). Diversity is all you need: Learning skills without a reward function. *arXiv preprint arXiv:1802.06070*.

We thank the reviewer again for their comments.

REVIEWERS' COMMENTS

Reviewer #3 (Remarks to the Author):

3rd review of 'Learning Few-Shot Imitation as Cultural Transmission' by Avishkar Bhoopchand, et al., DeepMind, for Nature Communications

The paper is excellent and merits publication. It expands the state of the art on several fronts, and will serve as a very valuable contribution to the literature.

Minor suggestions:

p. 17 – comma after 'wayfinding'

p. 19 – 'Earlier, we argued that appropriate randomisation over experts may generate a selection process for knowledge'

– It would be helpful to the reader to know what section this was in to facilitate looking back to that section.

Also, for reasons outlined in my initial review (in the paragraph on 'selection') I would suggest replacing the phrase 'generate a selection process for knowledge' with 'facilitate knowledge evolution.' (Or a footnote to clarify that you are not necessarily using the word 'selection' in its scientific sense, as in 'natural selection'). Though I don't insist on this as the paper is now much clearer on the issue.

p. 19 – 'One could imagine comparing the various ablated models of MEDAL-ADR with the behaviour of children at different stages of ontogeny, or with the behaviour of non-human animals,'

– the authors could add 'or with hominids at early stages of our evolution', as was done in the computational model in this paper:

Gabora, L. & Smith, C. (2018). Two cognitive transitions underlying the capacity for cultural evolution. *Journal of Anthropological Sciences*, 96, 27-52. doi: 10.4436/jass.96008
[<https://arxiv.org/abs/1811.10431>]

Dr. Liane Gabora

Reviewer #3 (Remarks to the Author):

The paper is excellent and merits publication. It expands the state of the art on several fronts, and will serve as a very valuable contribution to the literature.

Minor suggestions:

p. 17 – comma after ‘wayfinding’

Thank you, we have inserted this.

p. 19 – ‘Earlier, we argued that appropriate randomisation over experts may generate a selection process for knowledge’

– It would be helpful to the reader to know what section this was in to facilitate looking back to that section.

Also, for reasons outlined in my initial review (in the paragraph on ‘selection’) I would suggest replacing the phrase ‘generate a selection process for knowledge’ with ‘facilitate knowledge evolution.’ (Or a footnote to clarify that you are not necessarily using the word ‘selection’ in its scientific sense, as in ‘natural selection’). Though I don’t insist on this as the paper is now much clearer on the issue.

We have amended this to read:

"Earlier in the discussion, we argued that appropriate randomisation over experts may generate selective social learning."

p. 19 – ‘One could imagine comparing the various ablated models of MEDAL-ADR with the behaviour of children at different stages of ontogeny, or with the behaviour of non-human animals,’

– the authors could add ‘or with hominids at early stages of our evolution’, as was done in the computational model in this paper:

Gabora, L. & Smith, C. (2018). Two cognitive transitions underlying the capacity for cultural evolution. *Journal of Anthropological Sciences*, 96, 27-52. doi: 10.4436/jass.96008

[\[https://arxiv.org/abs/1811.10431\]](https://arxiv.org/abs/1811.10431)

Thank you for the suggestion. We have decided not to include the suggested clause, since the comparison we were envisaging would involve direct experimental work involving animals or human children, which is clearly not possible for early hominids. To make this clear we have edited the clause as follows, to include the word "experiments":

"One could imagine experiments comparing the various ablated models of MEDAL-ADR with the behaviour of children at different stages of ontogeny, or with the behaviour of non-human animals,"

We thank the reviewer again for their comments.